# Primary succession and its driving variables – a sphere-spanning approach applied in proglacial areas in the upper Martell Valley (Eastern Italian Alps)

Katharina Ramskogler*, Institute for Alpine Environment, Eurac Research, Drususallee /Viale Druso 1, Bozen/Bolzano, 39100, Italy, katharina.ramskolger@eurac.edu; and Department of Botany, University of Innsbruck, Sternwartestraße 15, Innsbruck, 6020, Austria, katharina.ramskogler@student.uibk.ac.at

Bettina Knoflach*, Department of Geography, University of Innsbruck, Innrain 52, Innsbruck, 6020, Austria, Bettina.Knoflach@uibk.ac.at

Bernhard Elsner, Kompass-Karten GmbH, Karl-Kapferer-Straße 5, Innsbruck, 6020, Austria, bernhard.elsner@kompass.at

Brigitta Erschbamer, General-Feuerstein -Str. 24, Innsbruck, 6020, Austria, Brigitta.Erschbamer@uibk.ac.at

Florian Haas, Physical Geography, Catholic University of Eichstätt-Ingolstadt, Osten 14, Eichstätt, 85072, Germany, florian.haas@ku.de

Tobias Heckmann, Physical Geography, Catholic University of Eichstätt-Ingolstadt, Osten 18, Eichstätt, 85072, Germany, tobias.heckmann@ku.de

Florentin Hofmeister, Chair of Hydrology and River Bassin Management, Technical University of Munich, Arcisstr. 21, Munich, 80333, Germany, Florentin.hofmeister@tum.de

Livia Piermattei, Remote sensing Group, Research Unit Land Change Science, Swiss Federal Institute for Forest Snow and Landscape Research WSL, Zürcherstrasse 111, 8903, Birmensdorf, Switzerland, livia.piermattei@wsl.ch

Camillo Ressl, Department of Geodesy and Geoinformation, TU Wien, Wiedner Hauptstraße 8, Vienna, 1050, Austria, Camillo.Ressl@geotuwien.ac.at

Svenja Trautmann, Department of Geography, University of Innsbruck, Innrain 52, Innsbruck, 6020, Austria, svenja.trautmann@gmx.de

Michael H. Wimmer, Federal Office of Metrology and Surveying (BEV), Arltgasse 35, Vienna, 1020, Austria, Michael.Wimmer2@bev.gv.at

Clemens Geitner, Department of Geography, University of Innsbruck, Innrain 52, Innsbruck, 6020, Austria, clemens.geitner@uibk.ac.at

Johann Stötter, Department of Geography, University of Innsbruck, Innrain 52, Innsbruck, 6020, Austria, Hans.Stoetter@uibk.ac.at

Erich Tasser, Institute for Alpine Environment, Eurac Research, Drususallee /Viale Druso 1, Bozen/Bolzano, 39100, Italy, erich.tasser@eurac.edu

*These authors contributed equally to this work.
*Correspondence to*: Katharina Ramskogler (katharina.ramskogler@eurac.edu), Bettina Knoflach (bettina.knoflach@uibk.ac.at)

# Primary succession and its driving variables – a sphere-spanning approach applied in proglacial areas in the upper Martell Valley (Eastern Italian Alps)

Katharina Ramskogler[1,2]*, Bettina Knoflach[3]*, Bernhard Elsner[4], Brigitta Erschbamer[5], Florian Haas[6], Tobias Heckmann[6], Florentin Hofmeister[7], Livia Piermattei[8], Camillo Ressl[9], Svenja Trautmann[3], Michael H. Wimmer[10], Clemens Geitner[3], Johann Stötter[3], Erich Tasser[1]

[1]Institute for Alpine Environment, Eurac Research, Bozen/Bolzano, 39100, Italy
[2]Department of Botany, Universität Innsbruck, Innsbruck, 6020, Austria
[3]Department of Geography, Universität Innsbruck, Innsbruck, 6020, Austria
[4]Kompass-Karten GmbH, Innsbruck, 6020, Austria
[5]General-Feuerstein -Str. 24, Innsbruck, 6020, Austria
[6]Physical Geography, Catholic University of Eichstätt-Ingolstadt, Eichstätt, 85072, Germany
[7]Chair of Hydrology and River Bassin Management, Technical University of Munich, Munich, 80333, Germany,
[8]Remote sensing Group, Research Unit Land Change Science, Swiss Federal Institute for Forest Snow and Landscape Research WSL, Birmensdorf, 8903, Switzerland
[9]Department of Geodesy and Geoinformation, TU Wien, Vienna, 1050, Austria
[10]Federal Office of Metrology and Surveying (BEV), Vienna, 1020, Austria

*These authors contributed equally to this work.

*Correspondence to*: Katharina Ramskogler (katharina.ramskogler@eurac.edu), Bettina Knoflach (bettina.knoflach@uibk.ac.at)

**Abstract.** Climate change and the associated glacier retreat lead to considerable enlargement and alterations of the proglacial systems. The colonisation of plants in this ecosystem was found to be highly depending on terrain age, initial site conditions and geomorphic disturbances. Although the explanatory variables are generally well understood, there is little knowledge on their collinearities and resulting influence on proglacial primary succession. To develop a sphere-spanning understanding of vegetation development, a more interdisciplinary approach was adopted. In the proglacial areas of Fürkele-, Zufall-, and Langenferner (Martell Valley/Eastern Italian Alps), totally 65 plots of $5 \times 2$ m were installed to perform the vegetation analysis on vegetation cover, species number, and species composition. For each of those, 39 potential explanatory variables were collected, selected through an extensive literature review. To analyse and further avoid multicollinearities, 33 of the explanatory variables were clustered via Principal Component Analysis (PCA) to five components. Subsequently, Generalised Additive Models (GAM) were used to analyse the potential explanatory factors of primary succession. The results showed that primary succession patterns were highly related to the first component ('elevation and time'), the second component ('solar radiation'), and the third component ('soil chemistry') as well as the fifth component ('soil physics'), and landforms. In summary, the analysis of all explanatory variables together provides an overview of the most important influencing variables and their interactions, and thus a basis for the debate on future vegetation development in a changing climate.

## 1 Introduction

Due to climate change and the associated glacier retreat, proglacial areas, which are defined as landscapes that became deglaciated since the high stand glacier extent of the Little Ice Age (LIA, mid-19th century, e.g., Heckmann and Morche, 2019) and references therein), undergo considerable enlargement and changes due to geomorphic processes and also as a consequence of vegetation development as well as interaction between the different processes. The extent and rate of change in proglacial areas (e.g., primary succession) are influenced by different variables. According to the geoscientific concept of spheres (Sintubin, 2008; Stötter et al., 2014), these variables can be assigned to different spheres (Biosphere, Atmosphere, Cryosphere as part of the Hydrosphere, Relief sphere, Pedosphere, and Anthroposphere). Sintubin (2008) and Stötter et al. (2014) outlined that these spheres are interconnected and influence each other, to varying degrees, which has been supported by a large number of studies (e.g., Arnold et al., 1990; Kastens et al., 2009; Lin, 2010). The main interactions are highlighted in Figure 1. Primary succession in the proglacial area is profoundly affected by variations in temperature and solar radiation (Kaufmann, 2002; Raffl et al., 2006; Schumann et al., 2016). However, due to the interconnection of the spheres, which are all significantly modified by climate warming, proglacial vegetation development is further indirect influenced in diverse ways (Wojcik et al., 2021, and references therein). Atmospheric changes have strongly altered the temporal and spatial distribution of (sub)surface frost and ice occurrences (high mountain cryosphere) (Hock et al., 2019). As a result, the time-depending processes, in dependency of e.g., deglaciation and the encompassing site conditions and geomorphic processes are constantly modified by changes of the cryosphere (Wojcik et al., 2021 and references therein). In addition, the climate driven changes of the mountain cryosphere directly impact both, the hydrological system (Wehren et al., 2010) and the relief sphere (Beniston, 2006), leading to substantial variations of microclimatic and microtopographic patterns. All these factors in turn control the dynamics of soil formation and alteration, and they are closely related to the spatial variability of vegetation development and plant diversity (e.g., Wojcik et al., 2021). The distinct changing vegetation patterns in the proglacial area during landscape evolution, in turn, modify the microclimatic regime (i.e., temperature and humidity) due to changes in surface albedo and in evapotranspiration (Larcher, 1984) and increase the aggregate stability of soils and lead to changes in sediment fluxes. The mutual reaction between the stabilising effects of vegetation traits and geomorphic processes is investigated e.g. by Eichel et al. (2013, 2016, 2018) and presented as a conceptual model in Wojcik et al. (2021). Vegetation development is not only influenced by the processes and interactions of the different spheres, but it is also affected by biotic variables, e.g. plant-plant-interaction (Losapio et al., 2021) as well as dispersal ability and seed availability (Jones and del Moral, 2009; Erschbamer and Mayer, 2011). Concerning the composition of the different life forms (Kaufmann and Raffl, 2002; Fickert et al., 2017) it was only shown how it changed along the successional gradient. Hodkinson et al. (2003) showed that especially early colonising plant species often had ectomycorrhizal associations. Within literature it was often shown that microbial community or faunal composition is correlated with vegetation development (Alfredsen and Høiland, 2001; Albrecht et al., 2010; Junker et al., 2021). Furthermore, also the anthropogenic impact on the system has to be mentioned, not only by the effect of global climate change but also by livestock grazing and/or trampling, practiced up to the proglacial areas (Theurillat et al., 1998).

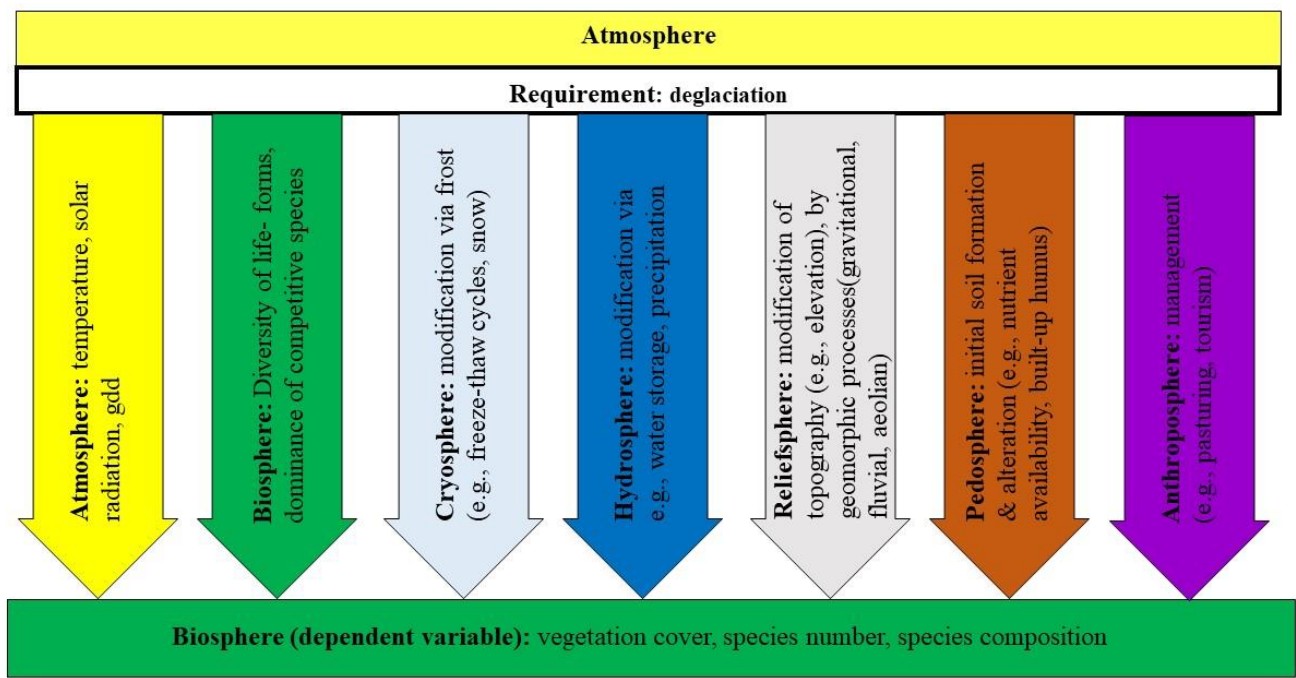

Figure 1: The main direct and indirect effects of atmospheric forcing on primary vegetation succession in proglacial systems (e.g., Hock et al., 2019; Wojcik et al., 2021).

The interaction of these variables leads to a spatially heterogenic pattern and development of proglacial areas which optically can be detected very clearly via the vegetation conditions on a small scale. In order to explain patterns of proglacial vegetation succession, it is necessary to include as many potential explanatory variables from different spheres as possible. The comprehension of these interaction processes will help to predict how proglacial areas will develop under future climate change. Scientists currently assume an increase in mean temperature with a stronger increase for the summer (Kotlarski et al., 2022), depending on the climate scenario, season, and region. Also a shift from solid to liquid precipitation due to higher temperature can be observed (Serquet et al., 2011; Kotlarski et al., 2022).

As summarised in Table 1, a large number of studies exist on primary succession in proglacial areas worldwide (e.g. European Alps: Raffl and Erschbamer, 2004; Fickert, 2020; Himalaya: Jiang et al., 2018; Andes: Llambí et al., 2021), although most of them only consider a few potential explanatory variables in their analyses. Although already Matthews (1992) recommended a multidisciplinary approach, to our knowledge only the study of Schumann et al. (2016) considers the influence of potential explanatory variables that covering all the spheres mentioned above. But even this study neglects some of the potential explanatory variables such as snow and (perennial) frost, which are expected to significantly change the patterns of primary succession in the face of climate change (Kaufmann and Raffl, 2002; Marcante et al., 2012).

In our study, we aimed to explain primary succession on proglacial areas in the Eastern Italian Alps by a sphere-spanning approach. We analysed the impact of as many potential explanatory variables as available, categorised according to the geoscientific concept of spheres. Our objectives were: (1) to conduct a comprehensive literature review on potential explanatory variables known to influence vegetation development in proglacial areas; (2) to investigate primary succession on proglacial areas in the upper Martell

Valley (Eastern Italian Alps) by recording total vegetation cover and plant species number. Therefore, we used many potential explanatory variables from literature as well as additional variables to test the following hypotheses: i) Many of the known potential explanatory variables are correlated and can be summarised into a few components. ii) It is not only single drivers used in literature that are decisive, but much more the interaction of all of them that influences vegetation cover. iii) Disturbances such as geomorphic disturbance and grazing/trampling reduce cover and species number, and thus also changes species composition. With the three

tested hypotheses we aim to provide a better understanding of primary succession for predicting future development.

## 2 Study area

The studied proglacial areas of the once united contiguous glaciers Fürkele-, Zufall-, and Langenferner are located within the Ortles-Cevedale group in the Upper Martell Valley (46.46 °N, 10.64 °E, Fig. 2a), Autonomous Province of Bozen/Bolzano, Italy. The study area extends from 2367 m above sea level (a.s.l.) to 2881 m a.s.l. and is NE-SW orientated. Totally 65 plots (Fig. 2c) were

sampled in 2019/2020 (used already for the analysis by Knoflach et al., 2021). They were located on the ground and lateral moraines of Fürkele- and Zufallferner, and at the lateral moraines of Langenferner along the elevation gradient. The study area is mainly characterised by chlorite-secerite leading micaschist consisting of alluvial and glacial deposits (Martin et al., 2009), which exhibit spatially very heterogeneous soil formations (Martin et al., 2009). Additionally, deposits of quartzite and marble can be found (Martin et al., 2009). The study area is located in the Central Alps within the tundra climate (ET) (Kottek et al., 2006) with a mean

annual daily mean air temperature of 2.9 °C (Station Zufritt; based on data from the 3PCLIM-project; source: www.3pclim.eu; accessed on 29.04.2023; Supplement, Figure S1a), and a mean annual sum of precipitation of 750 mm (Station Zufritt; based on data from the 3PCLIM-project; source: www.3pclim.eu; accessed on 29.04.2023; Supplement, Figure S1b) for the 30 years climate period 1981 to 2010 and 1980 to 2010, respectively.

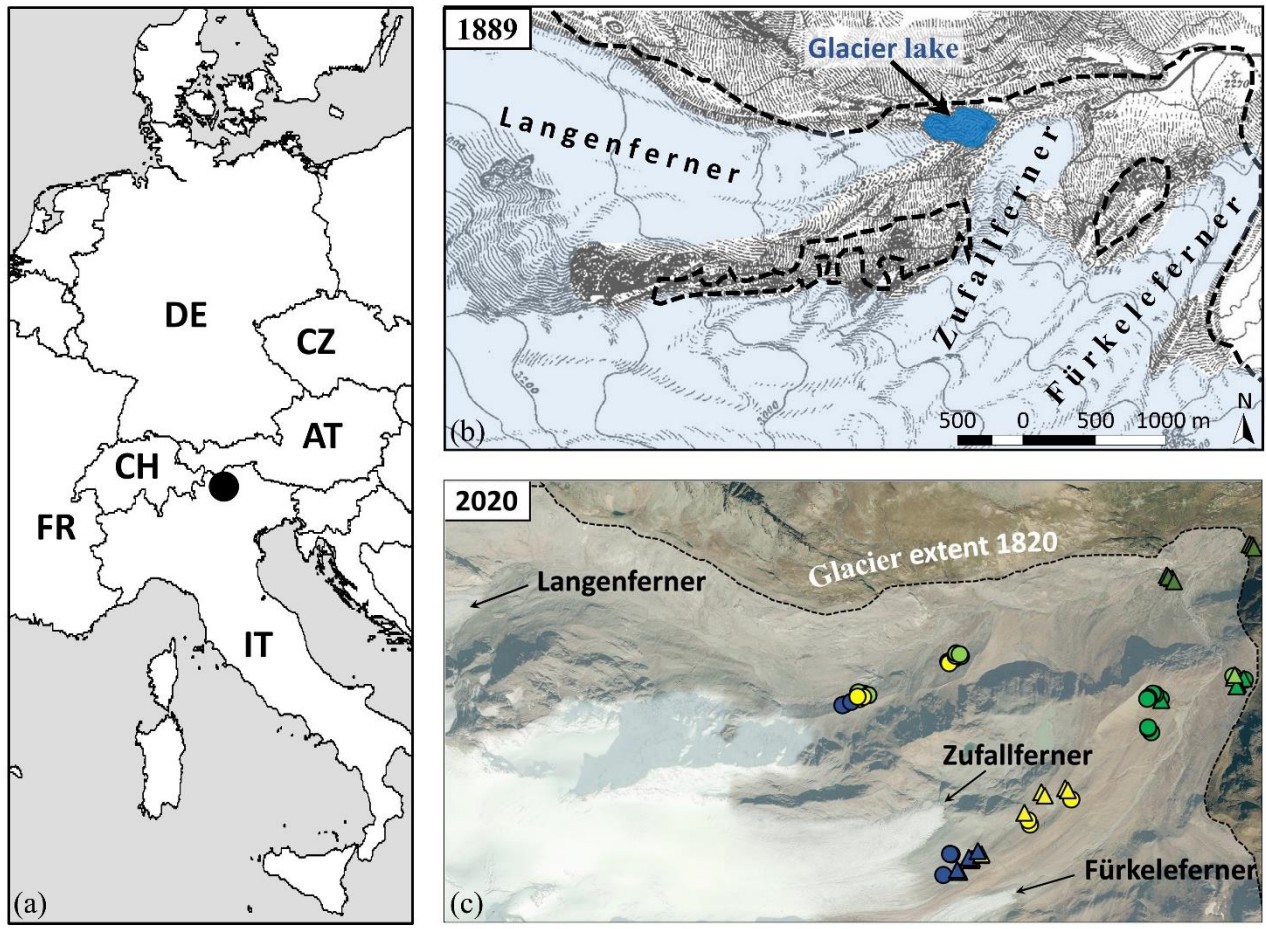


**Figure 2: (a) Study area and (b) glacier tongues of Fürkele-, Zufall-, and Langenferner in 1889 (Finsterwalder, 1890); modified) with the glacier extent around 1820 (Ivy-Ochs et al., 2009; Kinzl, 1932) (end of the LIA = black dashed line), (c) distribution of the vegetation plots in the proglacial area of Fürkele-, Zufall-, and Langenferner with the glacier extent around 1820 (Ivy-Ochs et al., 2009; Kinzl, 1932) (end of LIA = black dashed line; * = plots influenced by glacier lake outburst). Source of orthophoto: Autonomous Province of Bozen/Bolzano,**
**Italy 2020 (dots = geomorphologically disturbed plots, triangles = stable plots; blue = pioneer stage, yellow = early successional stage, yellowish green = late successional stage (snowbed community), light green = late successional stage (transition to grassland), and dark green = dwarf shrub stage).**

## 3 Material and methods

The study consists of two steps: in the first step, a literature review was performed for defining the potential explanatory variables.
In a second step, as many potential explanatory variables as possible were collected from the three proglacial areas of Fürkele-, Zufall- and Langenferner and implemented in a Principal Component Analysis (PCA) and Generalised Additive Models (GAM) as

well as a nonmetric multidimensional scaling to investigate their influence on primary succession. For the decision which biotic explanatory variables can be used an additional PCA was performed with the available variables.

## 3.1 Literature review: Definition of the potential explanatory variables

As a preselection of potential explanatory variables for primary succession (in particular, vegetation cover, species number, and species composition) in the proglacial area, we conducted a systematic review of existing literature (google scholar on 15/12/2021). To find articles which relate to vegetation analyses in proglacial areas, the search term had to contain vegetation- and glacier-related strings (intitle:vegetation OR intitle:plant OR intitle: succession AND intitle: glacier OR intitle:proglacial). Additionally, we added the string 'alps' to limit the search to alpine areas. From 176 publications, all non-scientific articles (only peer-reviewed journals

and books written in English or German language were considered), and those not focusing on primary vegetation succession in proglacial areas (e.g., studies on rock glaciers) were excluded manually. All remaining articles (n = 45) were considered (listed in the Supplement, Table S1). In total, 39 potential explanatory variables emerged from these articles (Fig. 3b). From these 39 potential explanatory variables we excluded variables only mentioned once or twice (e.g., wind exposure, snow depth, or soil type), except they could be relevant due to climate change, or variables not available for this study (e.g., soil depth, soil colour or soil texture).

For example, it can be assumed that snow depth is correlated with the snow free gdd, due to later melt out in places with higher snow cover (Unterholzner et al., 2022). We added new potentially explanatory variables as the Shannon-Index of the lifeforms and the relative cover of not-competitive species (Table 1). For the atmospheric variables we used additionally to the annual values also the one for the growing season (Table 1), which had not been used before. Thus, we ultimately used 38 potential explanatory variables and three dependent variables in our study (Table 1).


Table 1: Dependent and potential explanatory variables and assignment to the spheres as a result of the literature review, and details of the data sources and collection methods used in this study calculated for the entire year and the growing season (gs) (PCA = Principal Component Analysis, GAM = Generalised Additive Model). Some parameters were generated with the Water Flow and Balance Simulation Model WaSiM (Schulla and Jasper, 2019) and averaged for the years 2018/19 (*).

| Sphere | Dependent variables | Literature source | Data source | Methodical information | Used for the PCA and the GAM |
|---|---|---|---|---|---|
| **Biosphere** | Total vegetation cover (%) | 1, 2, 4-6, 9, 11, 12, 14-17, 20-22, 24, 25, 27, 29, 31, 35-39, 41-45 | Own field data from 2019/20 | | Yes |
| | Total species number (n) | 1, 3-7, 9, 11-13, 15, 18-22, 25-27, 29, 31, 34, 35, 37-42, 44, 45 | Own field data from 2019/20 | | Yes |
| | Successional stages | 1, 3, 4, 9, 12, 14, 20-24, 26, 28, 29, 32, 34-38, 40, 41, 44, 45 | (Knoflach et al., 2021) | Nonmetric Multidimensional Scaling (NMDS), Two Way INdicator SPecies ANalysis (TWINSPAN) | Yes |
| **Sphere** | **Potentially explanatory variables** | **Literature source** | **Data source** | **Methodical information** | |

| Biosphere | Shannon-Index of lifeforms | | (Landolt et al., 2010) | Calculation of the Shannon-index of relative cover of the different lifeforms | Yes |
|---|---|---|---|---|---|
| | Eveness of lifeforms | | (Landolt et al., 2010) | Calculation of eveness of the different lifeforms | No |
| | Ratio competitive/not-competitive species | | (Landolt et al., 2010) | csr-strategy types after Grime (1974) | No |
| | Amount competitive species | | (Landolt et al., 2010) | | No |
| | Amount not-competitive species | | (Landolt et al., 2010) | | No |
| | Relative cover competitive species | | (Landolt et al., 2010) | competitive species (2 or 3 times a 'c') | No |
| | Relative cover of not-competitive species | | (Landolt et al., 2010) | all other species = not-competitive species | Yes |
| Atmosphere | Solar radiation, annual (W m$^{-2}$ d$^{-1}$) * | [1, 6, 9, 10, 20, 26, 28, 33, 35, 37, 39] | WaSiM | corrected for inclination and aspect (Oke, 1987) | Yes |
| | Solar radiation, gs (W m$^{-2}$ d$^{-1}$) * | | WaSiM | corrected for inclination and aspect (Oke, 1987) | Yes |
| | Mean max. temperature, annual (°C) * | | WaSiM | mean daily temperature maxima, corrected for daily sunshine duration, aspect and local zenith angle (Schulla and Jasper, 2019) | Yes |
| | Mean max. temperature, gs (°C) * | | WaSiM | mean daily temperature maxima, corrected (see above) | Yes |
| | Mean temperature, annual (°C) * | [9, 13, 22] | WaSiM | mean daily temperature means, corrected (see above) | Yes |
| | Mean temperature, gs (°C) * | | WaSiM | mean daily temperature means, corrected (see above) | Yes |
| | Mean min. temperature, annual (°C) * | | WaSiM | mean daily temperature minima, corrected (see above) | Yes |
| | Mean min. temperature, gs (°C) * | | WaSiM | mean daily temperature minima, corrected (see above) | Yes |
| | Snow free gdd (n) * | [47] | WaSiM | calculated following Francon et al. (2021), Carlson et al. (2017), and Molau (1993) | Yes |
| | Wind exposure | [40] | | | No |
| Cryosphere | Years since deglaciation (n) | [1-13, 15-17, 19-45] | See supplement Table2 | | Yes |
| | Terrain age (n) | | Own data | years since glacial melt or last debris flow deposition or lake outburst | Yes |
| | Distance to glacier (m) | [4, 5, 13, 16, 18, 36, 43] | Orthophoto 2020 (©Autonome Provinz Bozen-Südtirol) | Derived with the 'near'-function in ArcGIS10.6® | Yes |
| | Snow free freeze-thaw days (n) * | | WaSiM | calculated following Schmidlin et al.( 1987) | Yes |
| | Days with snow cover (n) ' | [39] | WaSiM | calculated following Hofmeister et al. (2022) | Yes |
| | Snow depth | [45] | | | No |
| Hydrosphere | Topographic wetness index (TWI) | [1, 5, 10] | Derived from DTM (2019) | calculated in ArcGIS 10.6® | Yes |

| | | | | | |
|---|---|---|---|---|---|
| | | | (© Physical geography, KU Eichstätt-Ingolstadt) | | |
| | Flow accumulation (FA) | [10] | | | No (included in SPI) |
| | Moisture (site wetness) | [14] | | | No (not available for each plot) |
| | Sum precipitation, annual (mm d-1) | [1, 26] | WaSim | | Yes |
| | Sum precipitation, gs (mm d-1) | | WaSiM | | Yes |
| **Relief sphere** | Landforms | [5, 6, 8, 36, 40] | (Elsner et al., unpublished) | | Yes |
| | Geology | | | | No (only one site) |
| | Inclination (°) | [1, 4, 5, 10, 15, 20, 21, 26, 28, 33, 34, 37, 38, 40, 45] | Derived from DTM (2019) (© Physical Geography, KU Eichstätt-Ingolstadt) | calculated in ArcGIS 10.6® | Yes |
| | Aspect (northness/eastness) | [1, 4, 5, 9, 10, 15, 20, 21, 26, 33, 37, 38, 40, 44] | Derived from DTM (2019) ((© Physical Geography, KU Eichstätt-Ingolstadt) | aspect calculated in ArcGIS 10.6®, converted to northness/eastness according to Dial (2017) | Yes |
| | Elevation | [1, 2, 4, 6-12, 15, 20, 21, 26, 28, 32, 33, 35-38, 40, 45] | Derived from DTM (2019) ((© Physical Geography, KU Eichstätt-Ingolstadt) | | Yes |
| | Curvature | | Derived from DTM (2019) ((© Physical Geography, KU Eichstätt-Ingolstadt) | calculated in ArcGIS 10.6® | Yes |
| | Disturbance (Stream power index SPI) | | Derived from DTM (2019) ((© Physical Geography, KU Eichstätt-Ingolstadt) | calculated according to Florinsky (2017) in ArcGIS 10.6® | Yes |
| | Disturbance | [1, 6, 11, 20, 22, 36, 40, 43, 45] | Own field data from 2019/20 | Observations in the field (small gullies, visible instability) | No (as proxy used SPI) |
| **Pedosphere** | C | | | | |
| | pH | [1. 3. 5. 7. 9. 14. 16.19.20.26.29.31 33.38.] | Own field data from 2022 | In $CaCl_2$ (1:2.5), following VDLUFA (by Ecorecycling KG, Lana, Italy) | Yes |
| | Humus [%] | [1. 3.-5. 7. 9. 14. 16. 19. 20. 22. 29.1 31. 34. 39.] | Own field data from 2022 | UNI EN 15936 (TOC Analyzer) (by Ecorecycling KG, Lana, Italy) | Yes |

| | | | | | |
|---|---|---|---|---|---|
| | C.org [%] | 1. 3.-5. 7. 9. 14. 16. 19. 20. 22. 26. 29. 31. 33. 38. | Own field data | UNI EN 15936 (TOC-Analyzer) (by Ecorecycling KG, Lana, Italy) | Yes |
| | Cations | 5. 7. 14. 19. 20. | | | No |
| | Total nitrogen | 1. 4. 5. 7. 9. 14. 16. 19. 20. 22. 24. 29. 31. 34. | Own field data from 2022 | UNI EN 15936 (TOC Analyzer) (by Ecorecycling KG, Lana, Italy) | Yes |
| | P (CAL-P [mg P$_2$O$_5$/100g] | 5. 7. 19. 20. 24. | Own field data from 2022 | Using the Calcium-Acetat-Lactat method, following ÖNORM L 1087 (by Ecorecycling KG, Lana, Italy) | Yes |
| | Bare ground [%] | 1. 7. 20. 26. | Own field data from 2022 | Estimated in the field during vegetation surveys (1 class) | No |
| | Soil depth | 34, 37, 40, 45, | | | No |
| | Bulk density | 9, 19, 31. | | | No |
| | Litter depth | 34, 40, 45, | | | No |
| | Soil texture | 22, 40, | | | No |
| | K (CAL-K [mg K$_2$O/100g] | 5. | Own field data from 2022 | Using the Calcium-Acetat-Lactat method, following ÖNORM L 1087 (by Ecorecycling KG, Lana, Italy) | Yes |
| | Electric conductivity | 5. 14. | | | |
| | C/N | 20, 29, | Own field data from 2022 | UNI EN 15936 (TOC-Analyser) (by Ecorecycling KG, Lana, Italy) | Yes |
| | Sand [%] | 1. 5. 6. 14. 15. 21. 23.26. 31. 34. 36. 40, 41 | Own field data from 2022 | ÖNORM L1061-2, pipetting method (by Ecorecycling KG, Lana, Italy) | Yes |
| | Silt [%] | 1. 5. 6. 14. 15. 21.23. 26. 31. 34. 36. 40. 41. | Own field data from 2022 | ÖNORM L1061-2, pipetting method (by Ecorecycling KG, Lana, Italy) | Yes |
| | Clay [%] | 1. 5. 6. 14. 15. 21. 23. 26. 31. 34. 36. 40. 41. | Own field data from 2022 | ÖNORM L1061-2, pipetting method (by Ecorecycling KG, Lana, Italy) | Yes |
| | Soil type | 45, | | | No |
| | Soil skeleton | 1. 31. | | | No |
| | S | 1. 16. | | | No |
| | Soil colour | 40, | | | No |
| | Soil moisture, community weighted mean (m_w_F) | 6. 22. 26. 33. 38,-40, 45 | Own field data from 2019/20 | Gradient from very dry (1) to flooded/under water (5) (Landolt et al., 2010) | Yes |
| | Scree cover (%) | 1, 5, 6, 14, 15, 21, 23, 26, 31, 34, 36, 40, 41 | Own field data from 2019/20 | Estimated in the field during vegetation surveys (1 class) | Yes |
| **Anthroposphere** | Anthropogenic impact through grazing/trampling | 26 | Own field data from 2019/20 | Estimated in the field (Observation of livestock faces or trampling) max-standardisation (to take time into account) | Yes |

[1]Haselberger et al. (2021), [2]Knoflach et al. (2021), [3]Losapio et al. (2021), [4]Wei et al. (2021), [5]Wietrzyk-Pelka et al. (2021), [6]Wojcik et al. (2021), [7]Llambí et al. (2021), [8]Bayle (2020), [9]Fickert (2020), [10]Lambert et al. (2020), [11]Eichel (2019), [12]Fischer et al. (2019), [13]Franzén et al. (2019), [14]Szymański et al. (2019), [15]Fickert and Grüninger (2018), [16]Wietrzyk et al. (2018), [17]Mazhar et al. (2018), [18]Cazzolla Gatti et al. (2018), [19]Jiang et al. (2018), [20]D'Amico et al. (2017), [21]Fickert (2017), [22]Fickert et al. (2017), [23]Sitzia et al. (2017), [24]Göransson et al. (2016), [25]Erschbamer and

Caccianiga (2016), [26]Schumann et al. (2016), [27]Tampucci et al. (2017), [28]Carlson et al. (2014), [29]D'Amico et al. (2014), [30]Fischer (2013), [31]Burga et al. (2010), [32]Robbins and Matthews (2010), [33]Jones and del Moral( 2009), [34]Dolezal et al. (2008), [35]Raffl et al. (2006), [36]Mizuno (2005), [37]Caccianiga and Andreis (2004), [38]Raffl and Erschbamer (2004), [39]Kaufmann and Raffl (2002), [40]Andreis et al. (2001), [41]Burga (1999), [42]Frenot et al. (1998), [43]Mizuno (1998), [44]Vetaas (1994), [45]Matthews and Whittaker (1987)

## 3.2 Dependent variables: Vegetation sampling (Biosphere)

The vegetation surveys in the proglacial areas of Fürkele-, Zufall-, and Langenferner were carried out in July/August of 2019 and 2020 (n = 65) (Fig. 2c). For each of the study plots with a size of 5 m × 2 m, the total vegetation cover, all individual vascular species, and their cover (%) were recorded. The nomenclature of the species follows Fischer et al. (2008). Mosses and lichens were considered as own functional groups. According to the change in species composition along the chronosequence, Knoflach et al. (2021) discriminated four successional stages: (i) a pioneer stage, (ii) an early successional stage, (iii) a late successional stage with snowbed and grassland communities, and (iv) a dwarf shrub stage, by performing a Nonmetric MultiDimensional Scaling (NMDS) and a Two Way INdicator SPecies ANalysis (TWINSPAN).

## 3.3 Potential explanatory variables

### 3.3.1 Atmosphere

Based on the meteorological data from different weather stations (Supplement, Table S2), distribution maps (25 m × 25 m) of daily temperature (maxima, mean, and minima), and topographically corrected solar radiation were generated by the fully distributed Water Flow and Balance Simulation Model (WaSiM) (Schulla and Jasper, 2019). We spatially interpolated the meteorological variables temperature, wind speed, and humidity with an elevation dependent regression algorithm. Precipitation and solar radiation were interpolated with an inverse divergence weighting (IDW) method. The model setup and parameterisation published by Hofmeister et al. (2022) was used. The beginning and the end of the growing season were defined, following Bishop and Bishop (2014), by the mean temperature of 0 °C for four consecutive days (above 0 °C: start of the season; below 0 °C: end of the season) and the simultaneous absence of snow cover. Following Molau (1993), Carlson et al. (2017), and Francon et al. (2021), the number of annual snow free growing degree days (gdd) was defined as number of days with a mean daily air temperature above 0 °C. This was chosen because, especially in high alpine environments, germination and growth are possible at temperatures just above 0 °C (Kost, 2014). The calculation was done for the years 2018 and 2019 as the youngest plots were at least deglaciated since 2018. All atmosphere-related variables are listed in Table 1.

### 3.3.2 Biosphere

To take also the biosphere into account we used the Shannon-Index of the lifeforms, calculated from the relative cover of the different lifeforms. For the different lifeforms the values were extracted from Landolt et al. (2010). The csr-strategy types (Grime, 1974) were also extracted from Landolt et al. (2010). The species were grouped to competitive species with two or three 'c' and

not-competitive species (all other species). We also included the relative cover of the not-competitive vascular plant species (Table 1).

### 3.3.3 Cryosphere

In order to determine the temporal exposure of the study plot to non-glacial conditions, a high-resolution glacial reconstruction was
carried out based by using (i) hill shade maps created from Airborne Laser Scanning (ALS) digital elevation models, (ii) aerial images, (iii) historical maps, and (iv) field investigations (Supplement, Table S3). The primary succession of some experimental plots was restarted after glacier lake outburst floods (Fig. 2b). Historical documents show that "the Martell Valley was affected by water catastrophes in the past two years (1888 and 1889), which in their magnitude and peculiarity as well as the intital reason for their causes, are likely to attract the interest of wider circles. […] […] in an unusual place an exalted glacier snout had formed, from
which the water must in all probability have poured out. […]" (Finsterwalder, 1890: 21; translated from German by Ramskogler). Seven study plots (*, Fig. 2c) are located in this lower proglacial area which was affected by outburst floods of these ice-dammed lakes (Fig. 1b). Further outburst floods of the ice-dammed lakes were reported from 1891 (DOeAV, 1891) and 1895 (DOeAV, 1895). Since these plots were affected by these high impact disturbances and the succession was restarted, the actual terrain age (time zero of succession) was considered as a further variable in the analyses (Table 1). There is no significant difference in the soil
parameters of these plots and plots with similar age not affected by the glacier lake outburst (affected plots: for humus [%] 3.62 (±0.44) in comparison to similar not affected plots humus [%] 2.61 (±0.25)).

The parameter 'distance to the glacier front' was determined as the shortest distance from every single study plot to the glacier tongue using the 'near' function in ArcGIS 10.6®. The extent of the glacier tongues comes from the years when the according plots were surveyed. Information on the number of days with snow cover was derived from WaSiM (Schulla and Jasper, 2019) for the
years 2018/19 from Hofmeister et al. (2022). Snowmelt is simulated with the energy balance method, which computes the energy fluxes on the top of the snowpack. In addition, snow redistribution processes based on gravitational slides and wind were also considered in the simulation results. For the parametrisation of WaSiM (Schulla and Jasper, 2019) we used the same model configuration and parametrisation as recently published by (Hofmeister et al., 2022). The distinction between no snow and snow cover was defined by a threshold of 5 mm snow water equivalent which has been used in multiple studies (e.g., Brutel-Vuilmet et
al., 2013; Najafi et al., 2016; Conway et al., 2021; Thornton et al., 2021; Hofmeister et al., 2022). The number of snow free freeze-thaw days was defined following Schmidlin et al. (1987) as days with a maximum temperature $> 0$ °C and a minimum temperature $< -2.2$ °C.

### 3.3.4 Hydrosphere

The two hydrosphere-related variables were the precipitation and the Topographic Wetness Index (TWI). Based on the
meteorological data from different weather stations (Supplement, Table II), distribution maps (25 m × 25 m) of the daily precipitation sums were also generated with the WaSiM model (Schulla and Jasper, 2019). The calculation was done using the years 2018 and 2019 as the youngest plots were at least deglaciated since 2018 (Table 1). The TWI was determined in ArcGIS 10.6® using the

ALS derived Digital Terrain Model (DTM) acquired during a flight campaign (9[th] August 2019) with the VP1 (VuxSys LR) mobile laser scanner (Riegl.com) (Operator: Chair of Physical Geography, Catholic University of Eichstätt-Ingolstadt) and the calculation methods after Beven and Kirkby (1979).

### 3.3.5 Relief sphere

Topographical parameters (elevation, inclination, curvature, and aspect) were based also on the ALS derived DTM of 2019. As aspect is a circular variable it had to be transformed into the linear variables eastness and northness by computing the sine and cosine, respectively (Table 1).

A geomorphological landform map provided by Elsner et al. (unpublished) was utilised to assign the study plots to the categories 'ground moraine' (n = 42) 'lateral/end moraines' (n = 18), and 'other landforms' (n = 5) e.g., lacustrine plain  and active flood plain, active river channel, displaced landslide mass. In addition, the Stream Power Index (SPI) was calculated according to Florinsky (2017) to provide a proxy for geomorphic disturbance by flowing water. To verify this modelled parameter, the study plots were also classified in terms of disturbance estimated in the field: (i) more stable (n = 27) and (ii) disturbed plots (n = 38), based on observations of small gullies due to erosion or other visible instability (for more details see Supplement, Fig. S2; the mean SPI values of the two classes were significantly different with $p < 0.0001$).

### 3.3.6 Pedosphere

Soil analyses were performed on soil samples derived from three sampling points (0-10 cm soil depth) for each of the study plots except the ones without soil development at the steep lateral moraines. The samples were taken in August 2022. Only for soil moisture we used the community weighted mean (m_w_) of the Landolt indicator value for soil moisture (F) (Landolt et al., 2010) which was obtained based on the single species cover on the plot. The suitability of indicator values as proxies for soil parameters was described among others by e.g., Anschlag et al. (2017), Descombes et al. (2020), and Simon et al. (2020). Soil samples were air-dried for one week and sieved afterwards up to 2 mm. Afterwards the soil samples were analysed based on following methods: for pH - in $CaCl_2$ (1:2.5), following  VDLUFA; sand, silt, and clay were measured using the pipetting method according to the ÖNORM L1061-2;  humus [%], organic carbon (C.org) [%], C:N ratio, and total nitrogen [%] following UNI EN 15936 (with a TOC-Analyser);  plant-available phosphorous (CAL-P [mg $P_2O_5$/100g] and plant-available potassium (CAL-K [mg $K_2O$/100g] using the Calcium-Acetat-Lactat-method following ÖNORM L 1087. Finally, the estimated cover of coarse-grained material (scree cover) for each plot in the field was used as an additional independent variable (scree cover).

### 3.3.7Anthroposphere

Through conversations with local farmers, information on livestock grazing as the most important human influence on proglacial ecosystems in the study area was retrieved and added in the list of independent variables. Signs of livestock grazing and/or trampling were determined as dichotomous variable (yes/no) for each plot and described in the following as anthropogenic impact. To account

for time, we calculated the max-standardised grazing intensity based on the number of animals per time period starting in 1869 (Literature for the livestock number: Supplement, Table S4).

**3.4 Data analysis**

Since many of the environmental variables are closely correlated, a PCA with varimax rotation and using the Kaiser's criterion of 1 was performed to avoid multicollinearity. The 33 variables were reduced to 5 rotated components (RC1 – RC5). The factor scores of the rotated components (RC) were obtained from the PCA with the package psych (Revelle, 2021) in R (R Core Team, 2019) and calculated for each sampling plot. The variables relative cover of not-competitive species, mean snow free freeze-thaw days, eastness, and mean number of days with snow cover were separately used in the GAM as they did not load sufficiently in the components RC1 to RC5.


GAM were then run using the R package mgcv (V.1.8-34; Wood (2011)). The independent variables were the factor scores of the RCs derived from the PCA as well as the variables excluded from the PCA after the Kaiser-Meyer-Olkin (KMO)-test limit of 0.5 260 (SPI, TWI, northness, and m_w_F) and the variables with low loadings in the PCA (relative cover of not-competitive species, snow free freeze-thaw days, eastness, and days with snow cover). As we also wanted to explain the effect of grazing on vegetation, we also added the max-standardised grazing intensity additionally. Furthermore, the categorical variable (landforms) was added. Due to their non-linear relationship with the dependent variable for the variables RC2, and RC4 the non-parametric smooth function was applied in the GAM for vegetation cover. For the model of total species number, the smooth function was applied for RC1. The 265 smooth function uses a thin plate regression spline basis for each and automatically selects the effective degrees of freedom (Wood, 2022).

The GAM model for total vegetation cover was fitted using a beta distribution, as this is appropriate for dependent variables with values between 0 and 1 (Ferrari and Cribari-Neto, 2004). Therefore, total vegetation cover was converted to values between 0 and 1 using the formula,

Equ. (1):

$$y^{cov} = \frac{y*(n-1)+0.5}{n}$$

(1)

where $y^{cov}$ the transformed ratio of the vegetation cover, $y$ is the percentage of cover divided by 100, and $n$ is the number of plots (Smithson and Verkuilen, 2006).

The equation for the smooth function ($f$) is defined as,

Eq. (2):

$$f(x) = \sum_{j=1}^{k} \beta_j b_j(x1)$$

(2)

where $\beta_j$ is a coefficient, $b_j x(1)$ is a basis function using by default a thin plate regression spline using which are smooths without knots and optimal low ranked. (Wood, 2022)

The equation of the final GAM for the analysis of the total vegetation cover was,

Eq. (3):

$$g(E[y^{cov}]) = \beta_0 + \beta_1 * RC1 + f_1 * RC2 + \beta_2 * RC3 + f_2 * RC4 + \beta_3 * RC5 + \beta_4 * SPI + \beta_5 * TWI + \beta_5 * Northness + \beta_6 * Standardised\ grazing + \beta_7 * Eastness + \beta_8 * Snow\ free\ freeze - thaw\ days + \beta_9 * Days\ with\ snow\ cover + \beta_{10} * Realtive\ cover\ ncomp + \beta_{11} * m\_w\_F + \beta_{12} * Landform \quad (3)$$

where $g()$ is the link function, $E$ the expectation, $y^{cov}$ the transformed ratio of the vegetation cover, $\beta_0$ the overall intercept, $\beta_i$ the intercept, and $f_i$ the smooth function.

For the analysis of the species number, the GAM model was fitted using a Poisson distribution. The equation for the final model for the analysis of the species number was,

Eq. (4):

$$g(E[y^{num}]) = \beta_0 + f_1 * RC1 + \beta_1 * RC2 + \beta_2 * RC3 + \beta_3 * RC4 + \beta_4 * RC5 + \beta_5 * SPI + \beta_6 * TWI + \beta_7 * Northness + \beta_8 * Standardised\ grazing + \beta_9 * Eastness + \beta_{10} * Snow\ free\ freeze - thaw\ days + \beta_{11} * Days\ with\ snow\ cover + \beta_{12} * Realtive\ cover\ ncomp + \beta_{13} * m\_w\_F + \beta_{14} * Landforms \quad (4)$$

where $g()$ is the link function, $E$ the expectation, $y^{num}$ the species number, $\beta_0$ the overall intercept, $\beta_i$ the intercept, and $f_i$ the smooth function.

Following Schröder and Reineking (2004) and Dormann (2011) the residuals of the models were tested for spatial autocorrelation using the Moran's I in the R package DHARMa (Hartig, 2022).

To analyse the influence of environmental variables on species composition, a NMDS was performed using the package vegan version 2.5.6 (Oksanen et al., 2020) based on square root transformation of the in-situ observations and Bray-Curtis dissimilarity. Plots without vegetation (n = 2) had to be excluded from the analysis. The environmental variables were fitted to show their influence on the species composition.

Furthermore, we calculated the relative cover of each species in each plot, and we defined characteristic species for each successional stage as species with > 4 % relative cover and minimum 4 % higher relative cover than in the other successional stages.

## 4 Results

### 4.1 Literature review: Definition of the dependent and potential explanatory variables

The most frequently analysed vegetation-related, dependent variables (biosphere) in the literature were total species number (n = 31), total species cover (n = 30), and species composition (n = 24; Fig. 3a). Significantly less studies addressed the cover of individual species (n = 2) (e.g., Wietrzyk et al., 2018), functional groups (n = 2) (e.g., Kaufmann and Raffl, 2002), individual strategy types (n = 2) (e.g., Fickert et al., 2017), or the development of the Normalised Difference Vegetation Index (NDVI) (n = 5) (e.g.,

Fischer et al., 2019; Lambert et al., 2020; Knoflach et al., 2021). In order to ensure a solid basis for comparing our results, we limited our analyses to the most frequently used dependent variables total vegetation cover, species number, and species composition for our further analyses.

The potentially explanatory variables derived from literature research were assigned to the different spheres, following the geoscientific concept of spheres (Sintubin, 2008; Stötter et al., 2014): atmosphere (n = 5), cryosphere (n = 4), hydrosphere (n = 4), relief sphere (n = 7), pedosphere (n = 18), and anthroposphere (n = 1) (Fig. 2b). In total, 39 potential explanatory (independent) variables were mentioned in the 45 articles considered (Fig. 3b). Most studies included solar radiation (n = 11) (e.g., Caccianiga and Andreis, 2004; D'Amico et al., 2017; Fickert, 2020; Lambert et al., 2020), years since deglaciation (n = 43) (e.g., Fickert and Grüninger, 2018; Jiang et al., 2018; Franzén et al., 2019; Knoflach et al., 2021; Wei et al., 2021), soil moisture (n = 9) (e.g., Matthews and Whittaker, 1987; Andreis et al., 2001; Raffl and Erschbamer, 2004; Szymański et al., 2019; Fickert, 2020), elevation (n = 22) (e.g., Carlson et al., 2014; Schumann et al., 2016; Llambí et al., 2021; Wei et al., 2021), inclination (n = 15) (e.g., D'Amico et al., 2017; Haselberger et al., 2021; Wietrzyk-Pelka et al., 2021), aspect (n = 14) (e.g., Caccianiga and Andreis, 2004; Raffl and Erschbamer, 2004; Fickert, 2017; Lambert et al., 2020; Wietrzyk-Pelka et al., 2021), and disturbance (n = 9) (e.g., Matthews and Whittaker, 1987; Andreis et al., 2001; Eichel, 2019) as well as soil-related parameters such as organic carbon (n = 16) (e.g., D'Amico et al., 2014; Jiang et al., 2018; Wietrzyk et al., 2018; Losapio et al., 2021), nitrogen (n = 14) (e.g., Raffl et al., 2006; Burga et al., 2010; Fickert, 2020; Wei et al., 2021), and pH (n = 15) (e.g., Raffl et al., 2006; Burga et al., 2010; D'Amico et al., 2017; Fickert, 2020; Wietrzyk-Pelka et al., 2021) (Fig. 3b). Variables that received the least attention, with only one mention, include annual temperature (Tampucci et al., 2017) and summer temperature (June/July/August) (Knoflach et al., 2021), wind exposure (Andreis et al., 2001), duration of snow cover (Kaufmann and Raffl, 2002), snow depth (Matthews and Whittaker, 1987), flow accumulation as a proxy for areas where water accumulates (Lambert et al., 2020), soil type (Matthews and Whittaker, 1987), soil colour (Andreis et al., 2001), and anthropogenic impact (Schumann et al., 2016) (Fig. 3b).

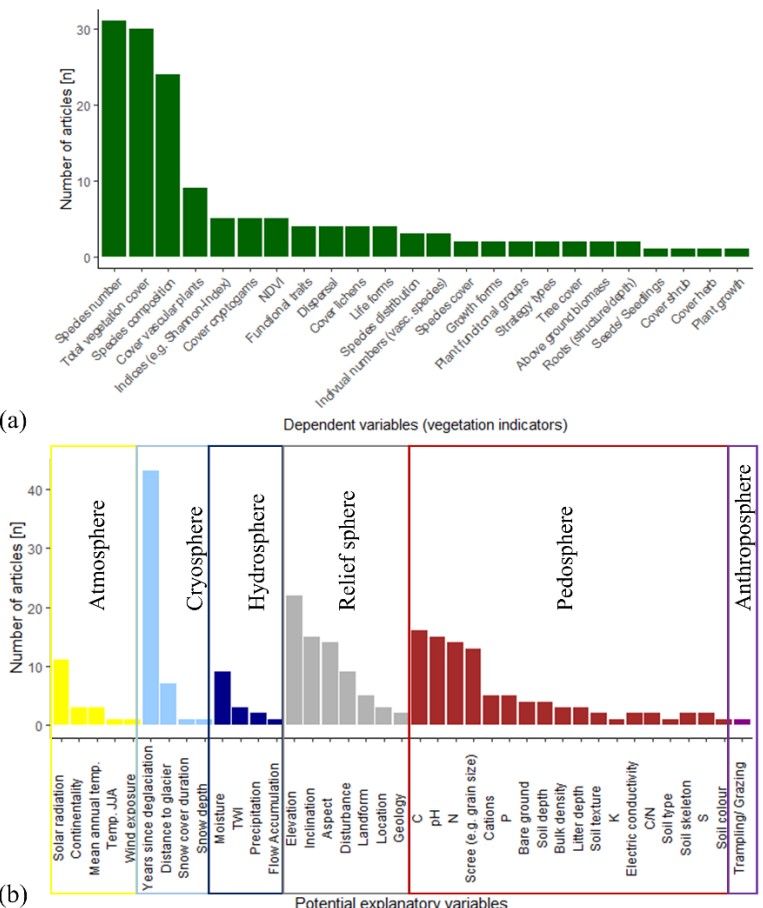

**Figure 3: Results of the literature research. Frequency of mentioned (a) vegetation indicators (dependent variables), and (b) potential explanatory variables. Colours indicate to which sphere the variable has been assigned.**

Although the potential explanatory variable temperature was considered less in literature, it was included in this analysis as it is important in times of changing climate. Furthermore, we decided to include the atmospheric variables (solar radiation for the growing season (gs), max. temperature, min. temperature, snow free gdd, and the temperature values for gs). Kaufmann and Raffl (2002) and Kreyling et al. (2007), for example, showed the relevant impact of the number of days with snow cover and the snow free freeze-thaw days on vegetation. Therefore, we included also these two variables although they were only mentioned once or even not found in our literature review. For the hydrosphere, this study included precipitation (annually and for gs) as they are also relevant due to climate change. We also used measured soil parameters, such as pH, content of sand, silt, clay, organic carbon, humus, total nitrogen, plant-available phosphorus, and potassium as well as the C:N ratio. To complete the spheres, we also used the Shannon-Index of lifeforms and the relative cover of not-competitive species. Thus, 39 potential explanatory variables remained for the analyses: two variables were assigned to the biosphere, nine variables to the atmosphere, five to the cryosphere, two to the

hydrosphere, seven to the relief sphere, eleven to the pedosphere, and one to the anthroposphere (Table 1). For geomorphic disturbance, only the SPI was used for the analyses (Table 1).

### 4.1.1 Reduction of potential explanatory variables to components

The variables relative cover of not-competitive species, mean snow free freeze-thaw days, eastness, and mean number of days with snow cover did not load sufficiently in the components RC1 to RC5. Thus, twenty-nine potential explanatory variables were reduced to five components. The five components (RC1 – RC5) explained 85 % of the variance. RC1 accounted for 35 % of the variance, RC2 for 16 %, RC3 for 13 %, RC4 for 10 %, and RC5 for 8 % respectively (Supplement, Table S5).

RC1 included, among others: years since deglaciation (0.90), elevation (- 0.91), annual temperature as well as temperature during the growing season, sum of precipitation (annual), and scree cover (- 0.75) (Table 2). Therefore, RC1 summarised key elevation-related climate parameters and variables connected with them, such as years since deglaciation, distance to glacier tongue or C:N ratio; it will be referred to as 'elevation and time'. RC2 included among others the solar radiation (0.89 and 0.88) and the snow free gdd (Table 2). This component will be designated as 'solar radiation'. RC3 was related to the content of organic carbon (0.87),
humus (0.87), total nitrogen (0.85), potassium (0.71), and curvature (0.54) (Table2). This component will be referred to as 'soil chemistry'. RC4 was negatively correlated with inclination (-0.79), and positively with pH (0.73), and sand (0.71; Table 2). Therefore, this component will be designated as 'inclination'. RC5 refers to silt (0.86), and clay (0.75) content (Table 2), hence this component will be referred to as 'soil physics'.

Table 2: Results of the PCA for our included driving variables in the proglacial area. The rotated factor loadings > 0.5 for the single components (RC) are given. Parameters which did not load sufficiently in any component (loading < 0.5) are highlighted in grey (gs = growing season, m_w_F = soil moisture).

| Parameters | RC1 'elevation and time' | RC2 'solar radiation' | RC3 'soil chemistry' | RC4 'inclination' | RC5 'soil physics' |
|---|---|---|---|---|---|
| Mean max. temperature annual | 0.94 | | | | |
| Mean max. temperature gs | 0.94 | | | | |
| Mean temperature gs | 0.92 | | | | |
| Mean min. temperature gs | 0.91 | | | | |
| Elevation | -0.91 | | | | |
| Mean temperature annual | 0.91 | | | | |
| Years since deglaciation | 0.90 | | | | |
| Mean min. temperature gs | 0.90 | | | | |
| Terrain age | 0.85 | | | | |
| Distance to glacier | 0.77 | -0.56 | | | |
| Scree cover | -0.75 | | | | |
| Sum precipitation annual | -0.67 | -0.57 | | | |
| Shannon-Index lifeforms | 0.62 | | | | |
| C/N-ratio | 0.62 | | | | |
| Sum solar radiation gs | | 0.89 | | | |
| Sum solar radiation annual | | 0.88 | | | |
| Snow free gdd | | 0.70 | | | |
| Mean precipitation gs | 0.53 | 0.57 | | | |
| Phosphorus | | 0.55 | | | |
| Organic carbon | | | 0.87 | | |
| Humus | | | 0.87 | | |
| Total nitrogen | | | 0.85 | -0.88 | |
| Potassium | | | 0.71 | 0.60 | 0.51 |
| Curvature | | | 0.54 | | |
| Inclination | | | | -0.79 | |
| pH | | | | 0.73 | 0.53 |
| Sand | | | | 0.71 | |
| Silt | | | | | 0.86 |

| | | |
|---|---|---|
| Clay | | 0.75 |


## 4.2. Effects of years since deglaciation, elevation, and climate on vegetation cover

In the GAM for total vegetation cover 90.4 % of the variance was explained by the model. The effect of the smoothed parameter, 'elevation and time' was highly significant (RC1, p < 0.0001), i.e., total vegetation cover increases with increasing RC1. But also, 'soil chemistry' and 'soil physics' had a significantly positive effect on vegetation cover (RC3, p < 0.0001; RC5, p < 0.0001), (Table

3a) as well as the smoothed parameters s(RC2), 'solar radiation' (p = 0.058) (Table 3b) increased vegetation cover. For the anthropogenic impact, the standardised grazing was not significant (Table 3). Regarding the different landforms, total vegetation cover was nearly significantly lower on lateral/end moraines (p = 0.063v) in comparison to ground moraines (Table 3a). Morans'I for the residuals of the model was not significant (p = 0.13), thus it was not necessary to account for spatial autocorrelation.

Table 3: (a) Effects of the analysed components and the additional potential explanatory variables (excluded from the PCA) on the total vegetation cover with the estimate, the standard error (SE), and the p-value (ncomp = not-competitive). (b) Effects of the smoothed terms with their approximate significance and their estimated degrees of freedom (edf), the reference degrees of freedom (Ref df) the Chi sq, and the p-value on the total vegetation cover. The significant variables are highlighted with bold numbers.

(a) Parametric coefficients.


| | Estimate | SE | p-value |
|---|---|---|---|
| (Intercept) | 0.018 | 2.40 | 0.99 |
| **RC1** | 1.562 | 0.171 | **<0.0001** |
| **RC3** | 0.424 | 0.123 | **<0.0001** |
| **RC5** | 0.584 | 0.130 | **<0.0001** |
| SPI | -0.00001 | -0.00003 | 0.742 |
| TWI | -0.023 | 0.068 | 0.732 |
| Northness | -0.238 | 0.175 | 0.174 |
| Standardised grazing | -0.105 | 0.624 | 0.866 |
| Eastness | -0.159 | 0.148 | 0.281 |
| Snow free freeze-thaw days | 0.102 | 0.065 | 0.115 |
| Number of days with snow cover | -0.006 | 0.005 | 0.301 |
| Relative cover ncomp species | -0.006 | 0.005 | 0.301 |
| m_w_F | 0.039 | 0.223 | 0.862 |
| **Lateral/end moraine** | -0.631 | 0.340 | **0.063** |

| | | | |
|---|---|---|---|
| Other landforms | -0,631 | 0.379 | 0.953 |

(b) Approximate significance of smooth terms.

| | edf | Ref df | Chi sq | p-value |
|---|---|---|---|---|
| **S(RC2)** | 2.342 | 2.923 | 7.294 | **0.05** |
| S(RC4) | 1.638 | 2.058 | 1.855 | 0.368 |


### 4.3 Important drivers for species number and composition

The utilised variables explained 89,4 % of the model variance. The proxy for geomorphic disturbance, SPI, had a weak negative effect (p = 0.074) (Table 4a). Max-standardised grazing had no significant (p = 0.869) effect on total species number (Table 4a). The model showed also a significantly positive effect of 'solar radiation' (RC2, p = 0.023), 'soil chemistry' (RC3, p = 0.045),
'inclination' (RC4, p = 0.0063), 'soil physics' (RC5, p < 0.0001), and eastness (p = 0.024) (Table 4a). On other landforms total species number had a weak positive effect (p = 0.094) in comparison to ground moraines; for lateral/end moraines the effect was not significant (Table 4a). Furthermore, there is a highly significant effect (p < 0.0001) for the smoothed term s(RC1), 'elevation and time'. This indicates that species numbers are higher at lower elevations and ice-free for longer times (RC1) (Table 4b). Morans'I for the residuals of the model was not significant (p = 0.229). Thus, it was not necessary to account for spatial
autocorrelation.

Table 4: (a) Effects of the analysed components and driving variables on the total species number with the estimate, standard error (SE), and the p-value (ncomp = not-competitive). (b) Effects of the smoothed term s(RC1) with their approximate significance and their estimated degrees of freedom (edf), the reference degrees of freedom (Ref df), the Chi sq, and the p-value on the total vegetation
cover. Significant variables in bold.
(a) Parametric coefficients.

| | Estimate | SE | p-value |
|---|---|---|---|
| (Intercept) | 2.022 | 0.892 | 0.023 |
| **RC2** | 0.131 | 0.058 | **0.023** |
| **RC3** | 0.105 | 0.052 | 0.045 |
| **RC4** | 0.171 | 0.063 | **0.006** |
| **RC5** | 0.218 | 0.037 | **<0.0001** |

| | | | |
|---|---|---|---|
| **SPI** | -0.00002 | 0.00001 | **0.074** |
| TWI | 0.008 | 0.025 | 0.723 |
| Northness | -0.002 | 0.057 | 0.975 |
| Max-standardised grazing | 0.035 | 0.211 | 0.870 |
| **Eastness** | -0.107 | 0.046 | **0.020** |
| Snow free freeze-thaw days | 0.046 | 0.029 | 0.111 |
| Days with snow cover | 0.001 | 0.003 | 0.801 |
| Relative cover ncomp species | -0.003 | 0.002 | 0.110 |
| m_w_F | 0.146 | 0.117 | 0.143 |
| Lateral/end moraine | 0.146 | 0.117 | 0.595 |
| **Other landforms** | 0.205 | 0.122 | **0.094** |

(b) Approximate significance of smooth term s(RC1).

| | edf | Ref df | Chi sq | p-value |
|---|---|---|---|---|
| **s(RC1)** | 5.473 | 6.613 | 93.420 | **< 0.0001** |

The NMDS (Fig. 4) clearly showed the relationships between the drivers and the successional stages, representing different species
compositions. The pioneer stage is positively correlated with scree cover, annual precipitation, and pH, as well as with snow free
freeze-thaw days, soil moisture, and elevation. Geomorphic disturbance correlates mainly with the early successional stages while
anthropogenic impact in max-standardised grazing by livestock is more common in the dwarf shrub stage. Solar radiation
corresponded with the early successional stage with the transition to grassland. The snowbed community occurs mainly on areas
with higher number of days with snow cover and SPI, and the transition to grassland on areas with more snow free gdd as well as
precipitation during the growing season, Shannon-Index of lifeforms, plant-available phosphorus. The dwarf shrub stage was highly
correlated with an increasing number of years since deglaciation, higher temperature values, higher values of soil humus content,
total nitrogen, and C:N ratio, and negatively with increasing elevation.


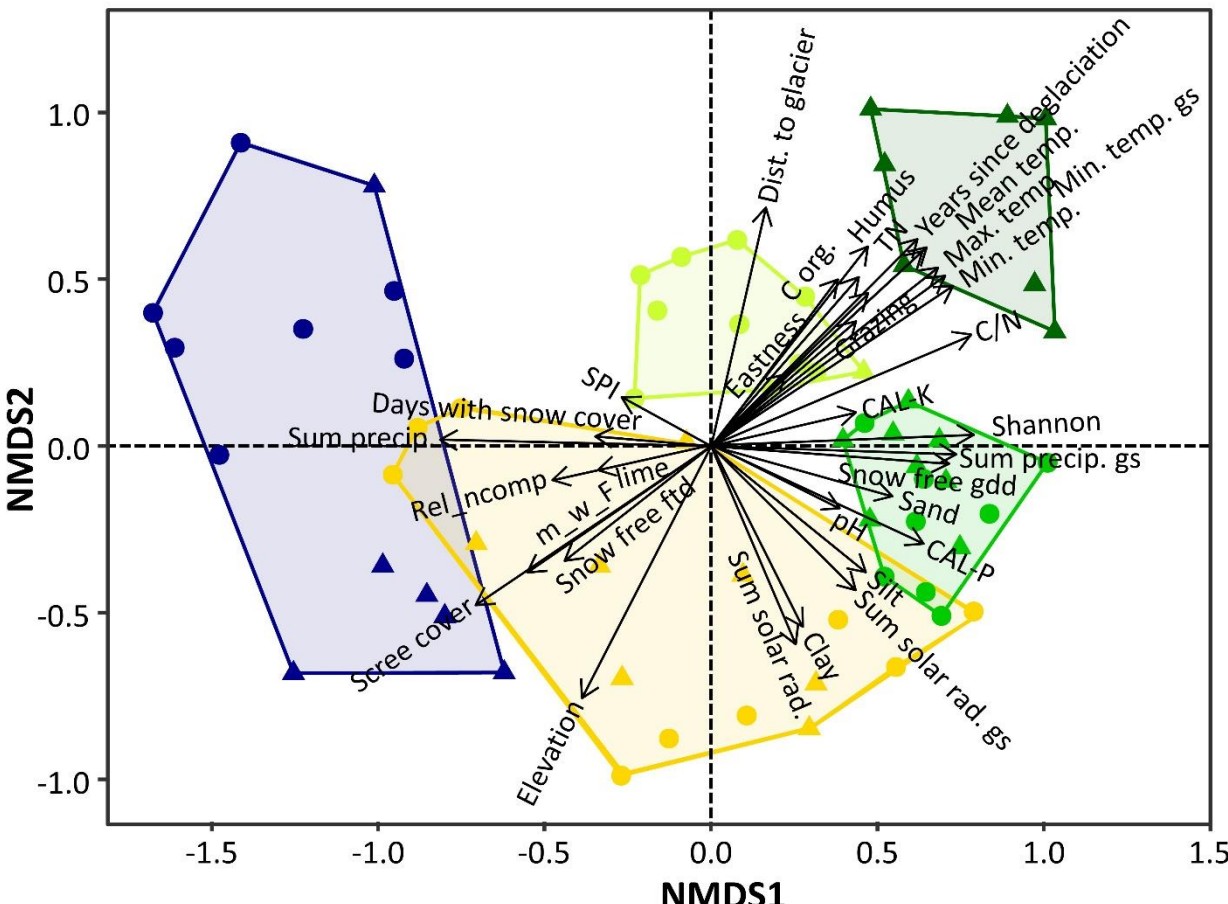

**Figure 4: Successional stages on the proglacial areas of Fürkele-, Zufall-, and Langenferner. The results are shown as a NMDS of the sampling plots. In the NMDS, the coloured dots (geomorphic disturbed), triangles (more stable), and polygons indicate the different species communities: blue = pioneer stage, yellow = early successional stage, yellowish green = late successional stage (snowbed community), light green = late successional stage (transition to grassland), and dark green = dwarf shrub stage. The length of the black arrows shows the high significant correlation between the driving variables on species composition (temp. = temperature, precip. = precipitation, dist. = distance, ftd = freeze-thaw days, gs = growing season, gdd = growing degree days, SPI = stream power index, m_w_F = community weighted mean of the indicator value for moisture, CAL-P = plant-available phosphorus, CAL-K = plant-available potassium, TN = total nitrogen, C.org = organic carbon, Shannon = Shannon-Index of the lifeforms, rel ncomp = relative cover not-competitive species).**


Species appearing first in the pioneer stage with the highest relative cover are mosses, and *Saxifraga oppositifolia* (Supplement, Figure S3, Table S6). In the early successional stage *Poa laxa*, mosses, *Polytrichum* sp., and *Salix helvetica* were the species with the highest relative cover (Supplement, Figure S3, Table S6). Amongst others, In the dwarf shrub stage the species with the highest

relative cover were *Rhododendron ferrugineum*, *Arctostaphylos uva-ursi*, *Cetraria islandica*, and *Empetrum hermaphroditum*
(Supplement, Figure S3, Table S6). Species which could be observed in all successional stages were *Cardamine resedifolia*, mosses, *Poa alpina*, *Polytrichum* sp., and *Racomitrium* sp. with different relative cover values (Supplement, Figure S3, Table S6). *S. aizoides* as well as *S. oppositifolia* disappeared after the early successional stage (Supplement, Figure S3, Table S6).

## 5 Discussion

Primary succession in proglacial areas is well studied, and many drivers of succession have been suggested in the literature.
However, most of the studies included only a few potential explanatory variables that in many cases do not cover all spheres (e.g., Knoflach et al., 2021; Fickert, 2020; Cazzolla Gatti et al., 2018; Robbins and Matthews, 2010; Mizuno, 2005; Andreis et al., 2001). This study stresses that many potential explanatory variables outlined in the literature are strongly correlated with each other and thus reflect the same phenomenon (statistically the same principal component). To prove this, we used as many known potential explanatory variables from as many spheres as possible in our approach. The set of 29 potential explanatory variables could be
reduced to five components due to multicollinearities. The first component 'elevation and time' is correlated with years since deglaciation, elevation, temperature, annual precipitation, scree cover, Shannon-Index of lifeforms, and C:N ratio. Several studies listed the variables summarised in our study in the component 'elevation and time' as important for the development of vegetation in the proglacial area: e.g., Andreis et al. (2001) and Wei et al. (2021) named years since deglaciation as well as elevation, Fickert (2020) concluded that years since deglaciation, elevation and temperature have an influence on vegetation development in the
proglacial area. Similarly, e.g., Schumann et al. (2016), Haselberger et al. (2021), and Wietrzyk-Pelka et al. (2021) stated that years since deglaciation, scree cover, soil reaction, or soil moisture have an impact on vegetation development. The higher C:N ratio with increasing terrain development is in line with the results shown by Khedim et al. (2021) for proglacial areas all over the world. Other studies (e.g, Schumann et al., 2016; Lambert et al., 2020) took solar radiation as separate variable into account whereas in our study this variable is highly correlated with snow free gdd, precipitation during the growing season, and plant-available phosphorus
in the component 'solar radiation'. The relevance of the snow free gdd for enhanced vegetation development was shown by Carlson et al. (2017) in the French Alps. Snow free gdd can also be related to snow depth as with higher snow depth the melt out will be later and thus we will have lower number of snow free gdd (Unterholzner et al., 2022). The component 'soil chemistry' is linked with organic carbon, humus content, total nitrogen, plant-available potassium, and curvature. The correlation between curvature and nutrients was shown for the high alpine environments by (Franz, 1979). Most of the previous studies analysed the effect of total
nitrogen and associated it directly with years since deglaciation, e.g., for the proglacial areas of Damma glacier (Göransson et al., 2016), the proglacial areas of Svalbard or the proglacial area of Urumqi glacier No. 1 in China (Wei et al., 2021). In contrast to these publications, our study revealed that nutrients are more dependent on the microtopography (curvature) and not only on time as also shown by Temme and Lange (2014) for three closely related proglacial areas in Switzerland. The component 'inclination' includes not only inclination but also pH, and content of sand. Finally, the component 'soil physics' is correlated with the amount of silt and

clay (Matthews, 1992). With our analyses we confirmed hypothesis (i) of variables collinearity and outlined an example of variable aggregation to fewer components.

## 5.1 Drivers for development of vegetation cover

For total vegetation cover we hypothesised that it is not a single drivers as used in literature that are decisive, but much more the
interaction of them. In previous studies, e.g., years since deglaciation (e.g., Matthews and Whittaker, 1987; Raffl et al., 2006; Erschbamer and Caccianiga, 2016; Schumann et al., 2016; Llambí et al., 2021), elevation (e.g., Raffl and Erschbamer, 2004; Lambert et al., 2020), mean annual temperature (e.g., Fickert et al., 2017; Franzén et al., 2019; Fickert, 2020), and annual precipitation (e.g., Schumann et al., 2016; Haselberger et al., 2021) resulted as essential drivers of vegetation development. Topography is influencing climatic variables as temperature and precipitation and thus the resulting microclimate affects vegetation cover (Scherrer and
Körner, 2011). Thus, microclimate is also partly represented by topography. In our study, we have now demonstrated that a series of other variables correlates with these three variables, and additional one can be jointly described by the components 'elevation and time', 'solar radiation', 'soil chemistry', 'inclination', and ('soil physics'. Thus, hypothesis (ii) that not a single drivers are decisive but the interaction of them can be confirmed. 'Elevation and time' includes scree cover for which the impact was also shown by e.g., Schumann et al. (2016) for the Eastern and Western Alps and C:N ratio for which the impact is also shown by
e.g.,D'Amico et al. (2017) in the proglacial area of the Verra Grande glacier in the North-Western Italian Alps.. In addition, we showed that also landforms constitute a significant impact. Rydgren et al. (2014) and Schumann et al. (2016) also suggested the positive effect of solar radiation on vegetation cover, which, by the way, correlated with the snow free gdd in our study. At the same time, solar radiation increases chemical weathering which is positively associated with enhanced rate of soil formation and thus higher plant-available phosphorus (Rech et al., 2001). Also, 'soil chemistry' is positive associated with vegetation cover as shown
for different proglacial areas, e.g., in Svalbard (Wietrzyk et al., 2018; Wietrzyk-Pelka et al., 2021) and 'soil physics' as shown by Burga et al. (2010) in the proglacial area of Morteratsch glacier in Switzerland.

The development of vegetation cover was found to differ weakly between landforms. We could observe lower cover on the lateral moraines in comparison to the ground moraines. The plots on the lateral moraines are the less stable ones. Another less investigated driver in previous studies was the impact of livestock grazing. With using the max-standardisation, we could not observe a significant
impact of livestock grazing/trampling on vegetation cover. Using grazing/trampling as categorical variable Schumann et al. (2016) showed a negative effect for cover of herbs, mosses, and lichens for proglacial areas in the Eastern and Western Alps.

## 5.2 Drivers for development of species number and species composition

For species number and species composition we hypothesised in particular that geomorphic and anthropogenic disturbances reduce
species number compared to undisturbed areas and thus also change species composition. Of course, the species number and composition are also influenced by the same components as the vegetation cover such as 'elevation and time', 'solar radiation', 'soil chemistry', 'inclination', and 'soil physics' as well as eastness. For example, Andreis et al. (2001) of 'soil chemistry' as well as 'soil physics' on species composition towards the later successional stages for the Western and Southern Italian Alps and Wietrzyk-

Pelka et al. (2021) for proglacial areas in Svalbard. With respect to our hypothesis (iii) on the effect of disturbances, the results are not clear. The stream power index (SPI) had a weak negative effect on species number, grazing and/or trampling showed no significant correlation. For species composition we could show a significant effect of SPI as well as grazing/trampling by livestock. Most of the studies in the literature came to the conclusion that geomorphic disturbance has an impact on species composition (D'Amico et al., 2017; Andreis et al., 2001; Matthews and Whittaker, 1987) but not on species number. Only Fickert and Grüninger, 2018) noticed that individual number and also species number decreased with disturbance in an early successional stage. Thereby, geomorphic disturbance controls succession not only by erosion but also by modification of the relief and relocation of sediments (Ballantyne, 2002). For species number Schumann et al. (2016) showed contrary to this study a significant positive effect of grazing, but they used grazing only as a categorial variable. The modification of the relief (microtopography, and grain size composition) can further lead to changes in microclimate, water and nutrient availability, and shading (Ballantyne, 2002). As grain-size has an impact on vegetation development (Burga et al., 2010), the relocation of sediments by geomorphic processes is important for the entire primary succession pathway. Andreis et al. (2001) showed a negative impact of disturbance on species composition but they included surface runoff as well as grazing and/or trampling in one factor. For snow cover duration an effect on species diversity was shown by Kaufmann and Raffl (2002), we could observe the effect of snow cover duration only for species composition.

As we had only one study area, we could not take lithology into account. But Mainetti et al. (2022) analysed two lithological different proglacial areas in the Gran Paradiso National Park and observed higher species number along the whole chronosequence in the siliceous area but lower vegetation cover, except for the first successional stage. The lower cover in their study might be due to higher elevation of the siliceous study area. Elevation of the study site in general matters for primary succession, especially for species composition, e.g. Burga et al. (2010) observed establishment of Swiss stone pine as well as European larch 15 to 31 years after deglaciation. Another example that elevation matters was shown in the study of Garbarino et al. (2010) observed germination of larch between 14 to 34 years after deglaciation with denser tree stands at the lower sites. Their study area was lower in comparison to our study site, thus also the density of trees was higher. They also showed that facilitation did not matter for establishment of larch seedlings at their sites (Garbarino et al., 2010). Erschbamer et al. (2008) mentioned that also the availability of safe sites is important for colonisation, especially in the early stages and that another limiting factor is limitation by dispersal. Another important factor for vegetation development in the proglacial area are the seed availability as well as the distance to and the size of the species pool (Erschbamer and Mayer, 2011). Also, plant-interactions (Erschbamer and Caccianiga, 2016; Losapio et al., 2021) are affecting vegetation development in these areas. For plant-arthropod interactions Kaufmann and Raffl (2002) showed that the first herbivorous families appeared when at least a bit vegetation was present. But not only herbivorous arthropods are affecting primary succession, also plant-pollinator interactions are influencing vegetation development during primary succession (Losapio et al., 2015).

Considering all variables used in this study and with a view to future development, we note that most of these variables are significantly influenced by advancing climate change. Not only regional climate is affecting primary succession but also microclimate (Wojcik et al. 2021, and references therein). For example, through the ongoing climate change, the growing season

will be prolongated and the growing conditions will be generally improved. Theurillat et al. (1998) suggested that per every increased degree in mean air temperature, the length of growing season will be enhanced by 16 to 17 days. Regarding precipitation, the ratio of snow relative to liquid precipitation shows a decreasing trend since the 1960s and 1970s in the Alps (Serquet et al., 2011), implicating less snowfall also in high elevations. Changes in snow cover can affect snow bed communities due to earlier melt or can also cause frost damages (Theurillat et al., 1998).

## 6 Conclusion

Our results clearly reveal that many drivers for primary succession found in literature refer to the same phenomenon (principal component), which reflects the interaction of spheres. The main drivers 'elevation and time', 'solar radiation', 'soil chemistry', and 'soil physics' as well as disturbance are influencing vegetation development during primary succession. It was demonstrated that vegetation cover, species number (diversity), and species composition are not always affected by the same set of drivers. The main difference is that species number is also affected by the component 'inclination'. However, in order to substantiate these results, further studies of a similar nature must follow, also in geologically different areas.

**Data availability**

All raw data can be provided by the corresponding authors upon request.

**Author Contributions:**

KR, BK, CG, BEr, JS and ET did the conceptualization- KR, B., BEl, FHo, FHa, TH, LP, CR, ST, MW, JS and ET did the data curation.; KR, BK, JS, and ET analysed the data and developed the methodology. CG, BEr, JS and ET did the funding acquisition and project administration. KR, BK, and ST conducted the field work and data collection. CG, BEr., JS, and E.T did the validation..;KR, BK, and ET did the visualisation. KR and BK wrote the manuscript draft. BEl, BEr, FHo., FHa., TH, LP, CR, ST, MW, CG, JS and ET reviewed and edited the manuscript. All authors have read and agreed to the published version of the manuscript.

**Conflicts of Interest**

The authors declare no conflict of interest.

**Acknowledgments**

We thank the Autonomous Province of Bozen/Bolzano, South Tyrol (Geoportal Alto Adige) for providing the aerial image of 2020. Thanks to the team of 'Die Historiker' for collecting the information about grazing. We thank the Stelvio/Stilfserjoch National Park for the permission to conduct the vegetation surveys and Iris Trenkwalder (2019) and Elias Nitz (2020) for their assistance during the field seasons. Alexander Wolff is thanked for his work in the completion and finalisation of the geomorphological map. Thanks to the people from the Marteller Hut, for the support during the field activities in 2019 and 2020. The soil analyses were performed by the external lab Ecorecycling KG, 39011 Lana (BZ), Italy.

**Funding**

The study was part of a cooperation between the SEHAG project ("Sensitivity of High Alpine Geosystems to Climate Change Since 1850"), financially supported by the Autonomous Province of Bozen/Bolzano, South Tyrol (IT-DFG 781607), the German Research Foundation (DFG) and the Austrian Science Fund (FWF) (grant numbers: HA 717 5740/10-1, HE 5747/6-1, ER 905/1-1, DI 639/5-1, CH 981/3-1, MA 6966/4-1, LA 4426/1-1 and 4062-N29) and the CryoSoil_TRANSFORM project, financially supported by the Austrian Academy of Sciences (ÖAW) as part of the Earth System Sciences (ESS) research program.

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
