# Peer review of "Primary succession and its driving variables – a sphere-spanning approach applied in proglacial areas in the upper Martell Valley (Eastern Italian Alps)"

_Biogeosciences, 2022_

## Author Response (AR1)

**Author's response and relevant changes**

**Reviewer 1**

**Common points**

**1** The paper aims to provide a "holistic approach" to investigate the ecological factors driving the primary succession on glacier forelands. On a case study in the Italian Alps, the authors tested the effects of a set of variables deriving from the previous literature in order to analyze their effects on plant cover and plant species composition.

> Response:
> Thank you for the positive evaluation of the study. We are confident that the revision of the manuscript helps to resolve the points of criticism.
>
> Relevant change:
> **Title:** Primary succession and its driving variables - a sphere-spanning approach applied in proglacial areas in the upper Martell Valley (Eastern Italian Alps)
> **New section added: 3.3.2 Biosphere** To take also the biosphere into account we used the Shannon-Index of the lifeforms, calculated from the relative cover of the different lifeforms. For the different lifeforms the values were extracted from Landolt et al. (2010). The csr-strategy types (Grime, 1974) were also extracted from Landolt et al. (2010). The species were grouped to competitive species with two or three 'c' and not-competitive species (all other species). We also included the relative cover of the not-competitive vascular plant species (Table1).

**2** The paper is interesting for the holistic approach, which takes into accounts the different "spheres", and for the nice literature review. Data are well collected, and the analytic methods are sound. The paper addresses scientific questions that are within the scopes of BG.

> Response:
> Thank you for confirming that the scientific questions of the manuscript are within the scopes of BG.

**3** However, some points need to be taken into account before publication. Sometimes the paper seems to be somewhat in the middle between a review and a research paper. The choice of the variables is somewhat constrained by their use in previous paper, while, in my opinion, the authors should have made their own choice. As a consequence, the hypotheses tested are a little bit trivial, and thus the paper does not tell anything really new.

> Response:
> Our intention behind this paper was indeed to consider all potential explanatory variables known from the literature for the important drivers (spheres) in the study. In doing so, we want to live up to our claim of presenting a study that is as holistic as possible. However, as the reviewer will have noticed, we have also included some additional indicators for certain drivers in the study (e.g., snow free freeze-thaw days, curvature). Above all, however, we wanted to combine all known explanatory variables and consider them in a joint analysis in order to work out the decisive variables. Thus, for example, combinations such as snow free freeze-thaw days, curvature, and temperature were used together as explanatory variables for the first time. However, we will take the hint very seriously in the revision and make the hypotheses more specific.
>
> Relevant change:
> **In introduction - hypotheses changed:** (i) Many of the known potential explanatory variables are correlated and can be summarised to a few numbers of components. ii) It is not only single drivers used in literature that are decisive, but much more the interaction of all of them. iii) Disturbances such as geomorphic disturbance and grazing/trampling reduce cover and species number and thus changes also species composition. With the three tested hypotheses we aim to provide a better understanding of primary succession for prediction of future development.

**4** The main problem, in my opinion, is that a true holistic approach cannot exclude completely the biotic factors as
drivers of the succession. Although the role of facilitation, competition etc have been part of the theory of primary
succession since its beginning, no biotic variable is considered as potential explanatory variables. Many papers
included some considered in the literature review (e.g. Losapio et al. 2021) showed that interspecific interactions are
a strong driver, as facilitation in the early phases and as competitive exclusion in the latter ones. It would have been
nice to take into account such topic, at least in the discussion. Also, propagule availability (under the form of distance
from potential sources) could have been taken into account, as well as the microclimatic effect of ice on the earliest
phases. I acknowledge that it is impossible to take into account everything, but given the "holistic" emphasis, a broader
consideration is expected. Also, a comparison with other forelands described in literature could be useful, as the
importance of some of the considered variables (e.g., temperature of the growing season) can be appreciated
comparing areas in different bioclimatic contexts.

Response:
Thank you for the positive evaluation of the study. We are confident that the revision of the manuscript helps
to resolve the points of criticism.
Relevant change:
**In section 3 Material and methods:** … For the decision which biotic explanatory variables can be used an
additional PCA was performed with the available variables.
**New section added: 3.3.2 Biosphere** To take also the biosphere into account we used the Shannon-Index of
the lifeforms, calculated from the relative cover of the different lifeforms. For the different lifeforms the
values were extracted from Landolt et al. (2010). The csr-strategy types (Grime, 1974) were also extracted
from Landolt et al. (2010). The species were grouped to competitive species with two or three 'c' and not-
competitive species (all other species). We also included the relative cover of the not-competitive vascular
plant species (Table1).

**5** Sometimes the discussion of the drivers is not very convincing: from one side, it does not provide ecological
hypothesis for the role of a variable (for example "south-eastness"- see below in the specific points); from the other,
it provides explications that sound a little bit too stretched. For example, to explain the role of slope (component 5) it
is said that its influence could be linked to the consequent soil properties, which lead to a lower C:N ratio (line 425):
but soil N content is one of the variables included in the independent component 4, so I would expect a connection
between the two.

Response:
The discussion will be adapted; some parts will be deepened, and others reduced.
Relevant change:
**Discussion** rewritten, due to new results after adding measured soil parameters.

**6** The first paragraph of the conclusions should be placed at the end of the discussion, as a separated paragraph dealing
with the potential effects of climate change. The conclusions should not treat a topic that has not been treated
elsewhere.

Response:
We will place the first paragraph of the conclusion at the end of the discussion.

**7** I suggest strengthenthe paper with a more robust description of the observed succession and its comparison with the
numerous case studies occurring in literature: does the succession imply addition and persistence or replacement? Can
we hypothesize from such features a role for biotic vs abiotic drivers? Which variables must be taken into account
and/or discussed?

Response:

We will show the observed successional sequence and its driving variables more decisively in the results by
indicating when which species appear and disappear, and which species persist over a long period of time.
Regarding the importance of biotic vs. abiotic factors, we will deepen the discussion.

Relevant change:
**In section 3.4. Data analysis last paragaraph:** Furthermore, we calculated the relative cover of each species
in each plot, and we defined characteristic species for each successional stage as species with > 4 % relative
cover and minimum 4 % higher relative cover than in the other successional stages.
**4.3. Important drivers for species number and composition**: Species appearing first in the pioneer stage
with the highest relative cover are mosses, and Saxifraga oppositifolia (Supplement, Figure S3, Table S6). In
the early successional stage Poa laxa, mosses, Polytrichum sp., and Salix helvetica were the species with the
highest relative cover (Supplement, Figure S3, Table S6). Amongst others, In the dwarf shrub stage the
species with the highest relative cover were Rhododendron ferrugineum, Arctostaphylos uva-ursi, Cetraria
islandica, and Empetrum hermaphroditum (Supplement, Figure S3, Table S6). Species which could be
observed in all successional stages were Cardamine resedifolia, mosses, Poa alpina, Polytrichum sp., and
Racomitrium sp. with different relative cover values (Supplement, Figure S3, Table S6). S. aizoides as well
as S. oppositifolia disappeared after the early successional stage (Supplement, Figure S3, Table S6).

**Specific points**

**1** 73-75 sentence unclear

Response:
This sentence will be rewritten.

**2** 79 Why giving only data for July? Yearly mean rainfall is important, as well as temperature at least for th whole
growing (or snow-free) season.

Response:
We will give the yearly mean rainfall, as well as the mean temperature for the whole growing season and the
mean annual temperature.

Relevant change:
**In section 2 Study area:** The study area is located in the Central Alps within the tundra climate (ET) (Kottek
et al., 2006) with a mean annual daily temperature of 2.9 °C (Station Zufritt; based on data from the 3PCLIM-
project; source: www.3pclim.eu; accessed on 29.04.2023; Supplement, Figure S1a), and a mean annual sum
of precipitation of 750 mm (Station Zufritt; based on data from the 3PCLIM-project; source: www.3pclim.eu;
accessed on 29.04.2023; Supplement, Figure S1b) for the 30-years climate period 1981 t0 2010 and 1980 to
2010, respectively.

**3** Line 124. I would not use the term "climax" for stages that are max 200 years old. Succession requires a much longer
time span to reach such stage (if it does)

Response:
Thank you for this hint. Climax will be replaced to dwarf shrub stage.

Relevant change:
… , and (iv) a dwarf shrub stage, by performing a Nonmetric MultiDimensional Scaling (NMDS) and a Two
Way INdicator SPecies ANalysis (TWINSPAN).

**4** 179 and following Data for the pedosphere are derived from Landolt's indices, which means that they derive from
plant species occurrence. Even if they are correlated with soil analyses performed on a small subsample, they are not very appropriate as explanatory variables for plant species composition, as they are not independent. I don't understand
why the interesting soil analyses weren't performed in all the plots and used as independent explanatory variables.

Response:
Due to financial constraints, we were unfortunately only able to sample a few sites in a first phase. In the
course of analysing the first results, however, we also realised that we needed concrete measurement data for
all sites. Therefore, we have now subsequently sampled all sites. We sampled all the vegetation plots in
summer 2022 and since end of december we have the results of the soil analyses from the lab. We will
integrate the measured soil data into the new analysis.

Relevant change:
**In section 3.3.6 Pedosphere:** Soil analyses were performed on soil samples derived from three sampling
points (0-10 cm soil depth) for each of the study plots except the ones without soil development at the steep
lateral moraines. The samples were taken in August 2022. Only for soil moisture we used the community
weighted mean (m_w_) of the Landolt indicator value for soil moisture (F) (Landolt et al., 2010) was obtained
based on the single species cover on the plot. The suitability of indicator values as proxies for soil parameters
was described among others by e.g., Anschlag et al. (2017), Descombes et al. (2020), and Simon et al. (2020).
Soil samples were air-dried for one week and sieved afterwards up to 2 mm. Afterwards the soil samples
were analysed based on following methods: for pH - in CaCl2 (1:2.5), following  VDLUFA; sand, silt, and
clay were measured using the pipetting method according to the ÖNORM L1061-2;  humus [%], organic
carbon (C.org) [%], C:N ratio, and total nitrogen [%] following UNI EN 15936 (with a TOC-Analyser);
plant-available phosphorous (CAL-P [mg P2O5/100g] and plant-available potassium (CAL-K [mg
K2O/100g] using the Calcium-Acetat-Lactat-method following ÖNORM L 1087.

**5** 200 and following. I agree that the human impact is of great importance for vegetation development. I wonder if
data collected only for the present time are appropriate for explaining the development of the succession. Are there
historical data about the past load of livestock somehow available, to check how representative is the present situation?

Response:
We have historical data of livestock density in this area. Instead of grazing (yes/no) we will use a max-
standardised value for grazing density.

Relevant change:
**In section 3.3.7 Anthroposphere:** ...To account for time, we calculated the max-standardised grazing
intensity based on the number of animals per time period starting in 1869 (Literature for the livestock number:
Supplement, Table S4).

**6** 295. How can be that PCA variables 3, 4 and 5 show an increasing explained variance?

Response:
Sorry for confusing you with our presentation of the components' results. Of course, you are right when you
point out that usually the explanatory share of the individual components decreases successively. This is of
course also the case with us. However, we wanted to rank the components in terms of the known effects. In
the revised version, however, we will now re-order the sequence according to the statistical output (so that
the declining explanatory share is again apparent).

Relevant change:
**In section 4.1.1 Reduction of potential explanatory variables - changes as described due to**
**implementing soil parameters:** The five components (RC1 – RC5) explained 83 85 % of the variance. RC1
accounted for 46 35 % of the variance, RC2 for 16 16 %, RC3 for 6 13 %, RC4 for 7 10 %, and RC5 for 8 %
respectively (Supplement, Table S5).

**7** 302. If RC2 is linked to solar radiation, what could be the ecological meaning of "south-eastness" of RC3? In the
discussion a possible explanation for this variable should be provided, particularly because in the conclusion, its
influence on vegetation cover is reported to be the main difference between vegetation cover and species richness.

Response:
The solar radiation and the exposure to the east/south do not correlate with each other so strongly that they are condensed into a single component. We therefore interpret that the exposure stands as a placeholder for additional processes/characteristics. Specifically, we are thinking above all of associated differences in precipitation patterns and the resulting soil water contents. Unfortunately, there is no high-resolution measurement data on this, so we cannot prove it. However, we will take up this topic in the discussion.

Relevant change:
**In section 4.1.1 Recution of potential explanatory variables - changes as described due to implementing soil parameters:** RC1 included, among others: years since deglaciation (0.90), elevation (-0.91), annual temperature as well as temperature during the growing season, sum of precipitation (annual), and scree cover (0.75) (Table 2). Therefore, RC1 summarised key elevation-related climate parameters and variables connected with them, such as years since deglaciation, distance to glacier tongue or C:N ratio.; it will be referred to as 'elevation' and time'. RC2 included among others the solar radiation (0.89 and 0.88) and the snow free gdd (Table 2). This component will bew designated as 'solar radiation'. RC3 was related to the content of organic carbon (0.87), humus (0.87), total nitrogen (0.85), potassium (0.71), and curvature (0.54) (Table 2). This component will be referred to as 'soil chemistry'. RC4 was negatively correlated with inclination (-0.79), and positively with pH (0.73), and sand (0.71) (Table 2). Therefore, this component will be designated as 'inclination'. RC5 refers to silt (0.86), and clay (0.75) content (Table 2), hence this component will be referred to as 'soil physics'.

**8** Line 411. I would not say that this hypothesis is not supported, as variable 1 is by far the most important. The fact that variable 1 includes other factors than those cited in the hypothesis is not meaningful: by definition, factors included in the same PCA axis cannot be disentangled, so the really significant ones could be just one or another of them, or all of them. So the hypothesis is very likely supported: the factors that were supposed to be the most significant are among those mainly contributing to the main variance. It's up to the authors' knowledge discuss which of them could or could not be important. For example, altitude and terrain age are correlated, as usual on glacier foreland: is the altitudinal interval big enough to represent an important factor? A comparison with similar intervals outside the LIA moraines could provide some insights.

Response:
Our explanations are too imprecise. The reviewer is absolutely right with the critism and we will adjust the text accordingly. As far as the comment to the ranks considered for the individual variables is concerned, our statements apply to comparable natural situations. However, in order to enable more general statements, we are currently in the process of investigating and analysing other glacier areas. In the current study, we have an elevation gradient of 500 m within the proglacial area - we think that the elevation interval is large enough to underline the importance of this variable. However, in order to enable more general statements, we are currently in the process of investigating and analysing other glacier areas with different elevation distributions.

Relevant change:
**Discussion** rewritten, due to new results after adding measured soil parameters.

**9** Maybe it's just me, but I don't find the figure 4 very clear.

Response:
Figure 4b (trampling/grazing) will change due to the changes described. In addition, we will reflect again on the current form.

Relevant change:
**In section 4.2 Effects of years since deglaciation, elevation, and climate on vegetation cover:** Figure 4 was deleted. / **In section 4.3 Important drivers for species number and composition:** Figure 5 was deleted.

**10** Line 431. I would not say that grazing "slowed down" the development of vegetation cover we see an effect on
plant cover and diversity, but it is unclear its role from the point of view of the succession.

Response:
We will take up this hint in order to deepen the discussion in this regard.
Relevant change:
**In section 5.1 Drivers for development of vegetation cover:** ...With using the max-standardisation, we
could not observe a significant impact of livestock grazing/trampling on vegetation cover. Using
grazing/trampling as categorical variable Schumann et al. (2016) showed a negative effect for cover of herbs,
mosses, and lichens for proglacial areas in the Eastern and Western Alps.
**11** Line 442. Replace "individuum" with "individual"

Response:
We will replace "individuum" with "individual

**Reviewer 2**

**Common points**

**1** The paper, after a necessary and important literature review, applies a multidisciplinary approach for studying
primary succession along a glacier foreland on European Alps. The aims of the paper fit well with the scopes of the
journal.

Response:
Thank you for the positive evaluation of the study. We are confident that the revision of the manuscript helps
to resolve the points of criticism. Also, thank you for confirming that the scientific questions of the
manuscript are within the scopes of BG.
**2** The review effort is a very important part of this paper and has a very international interest and application.

**3** The other part of the work (the application of the multidisciplinary approach to a case study) is, in my opinion, less
"holistic" than expected for different reasons:

Response:
The reviewer is of course correct in considering the implementation of the approach in our study. Due to
missing data (e.g., consumers) or missing variability (e.g., geology), the implementation does not correspond
to a holistic approach. However, we have predominantly applied the term to the approach of extracting all
potential drivers based on a literature review. However, to avoid creating the wrong impression in the title,
we will replace the term 'holistic' with a more appropriate term (probably 'sphere-spanning' or 'cross-
spheres').
Relevant change:
**Title:** Primary succession and its driving variables - a sphere-spanning approach applied in proglacial areas
in the upper Martell Valley (Eastern Italian Alps)
**3a** is a single case study, while it was already pointed out in literature that now, for having a new, innovative view
of proglacial habitat ("holistic"?), is necessary to have a synthesis of a wider spectrum of case studies. And it is
evident that some variables, at small scale, could not have a great importance, but, at bigger scale, are decisive (like
lithology). De facto, you compared two glacier forelands (not three; see comment below) of the same glacial site.
Some things, that you could not consider at small scale on a single site, should be considered in the discussion.

Response:

Thank you for this suggestion. We will consider the differences between different elevations and on different
geology as well as further topics which are important on a broader scale in the discussion.
Relevant change:
**In section discussion at the end:** As we had only one study area, we could not take lithology into account.
But Mainetti et al. (2022) analysed two lithological different proglacial areas in the Gran Paradiso National
Park and observed higher species number along the whole chronosequence in the siliceous area but lower
vegetation cover, except for the first successional stage. The lower cover in their study might be due to higher
elevation of the siliceous study area. Elevation of the study site in general matters for primary succession,
especially for species composition, e.g. Burga et al. (2010) observed establishment of Swiss stone pine as
well as European larch 15 to 31 years after deglaciation. Another example that elevation matters was shown
in the study of Garbarino et al. (2010) observed germination of larch between 14 to 34 years after deglaciation
with denser tree stands at the lower sites. They also showed that facilitation did not matter for establishment
of larch seedlings at their sites (Garbarino et al., 2010). Erschbamer et al. (2008) mentioned that also the
availability of safe sites is important for colonisation, especially in the early stages and that another limiting
factor is limitation by dispersal. Another important factor for vegetation development in the proglacial area
are the seed availability as well as the distance to and the size of the species pool (Erschbamer and Mayer,
2011). Also, plant-interactions (Erschbamer and Caccianiga, 2016; Losapio et al., 2021) are affecting
vegetation development in these areas. For plant-arthropod interactions Kaufmann and Raffl (2002) showed
that the first herbivorous families appeared when at least a bit vegetation was present. But not only
herbivorous arthropods are affecting primary succession, also plant-pollinator interactions are influencing
vegetation development during primary succession (Losapio et al., 2015).

**3b** among explanatory variables (mainly) only physical variables have been considered, excluding biological variables
like, for example, arthropods successions (see comments below) or plant interactions (see comment from the other
reviewer)

Response:
Thank you for the very interesting suggestion. We will include as further variables the proportion of
competing (ccc, ccs, ccr) and non-competing species (all other strategy types) in the analysis. In addition, we
will also consider the proportion of life forms (grasses, forbs, dwarf shrubs, lichens and mosses) as potential
explanatory variables. Finally, we will also elaborate the availability of propagation material, the
microclimatic effects of soil surface structure and the role of arthropods in the discussion.
Relevant change:
**3.3.2. Biosphere** To take also the biosphere into account we used the Shannon-Index of the lifeforms,
calculated from the relative cover of the different lifeforms. For the different lifeforms the values were
extracted from Landolt et al. (2010). The csr-strategy types (Grime, 1974) were also extracted from Landolt
et al. (2010). The species were grouped to competitive species with two or three 'c' and not-competitive
species (all other species). We also included the relative cover of the not-competitive vascular plant species
(Table1).

**4** These considerations do not give less importance to the work, that is very interesting, but suggest to valorise the fact
that a very detailed work on a single case study has be done focusing, with a great detail, on physical explanatory
variables. In my opinion, it could not be considered "holistic", it would be an error.

Response:
We will change the title according to what was said before.
Relevant change:
**Title:** Primary succession and its driving variables - a sphere-spanning approach applied in the proglacial
areas in the upper Martell Valley (Eastern Italian Alps)

**Specific points**

**1** Line 19: "proglacial areas …. undergo considerable enlargement and structural changes": this sentence should be explained. The enlargement is clear, Is less clear what do you mean with "structural Changes"

> Response:
> We will concretise this: We mean under structural changes - changes due to geomorphic processes and also as a consequence of vegetation development.
>
> Relevant change:
> We rewrote this sentence: ...undergo considerable enlargement and changes due to geomorphic processes and also as a consequence of vegetation development.

**2** Line 23: "which has been supported by a large number of studies". I think you should add some examples of studies

> Response:
> We will add some example studies here.
>
> Relevant change:
> Arnold et al. 1990, Kastens et al. 2009, Lin 2010

**3** Line 29: after "as a result" insert a comma.

> Response:
> We will insert the comma.

**4** Line 61: I would correct "Our objectives were: (1) We conducted…(2) We investigated.." in "Our objectives were: (1) to conduct…(2) to investigate.. etc".

> Response:
> Thank you, we will change it.
>
> Relevant change:
> Our objectives were: (1) to conduct a comprehensive literature review on potential explanatory variables known to influence vegetation development in proglacial areas, and (2) to investigate primary succession on proglacial areas in the upper Martell Valley (Eastern Italian Alps) by recording total vegetation cover and plant species number.

**5** Line 67: if your objective is also to test hypothesis.

> Response:
> We rewrite the sentence: Therefore, we used the from literature known potential explanatory variables and tested the following hypotheses:
>
> Relevant change:
> Therefore, we used the from literature known potential explanatory variables and tested the following hypotheses:

**6** Line 63: "three proglacial areas": from the map I see only two proglacial succession. I checked Knoflach et al. (2021) and I have seen that for the third proglacial area only lateral moraine has been sampled: in my opinion you could not consider this sampling on the third glacier foreland as a sampling of a proglacial succession. In addition, you compared proglacial areas of the same site, thus, they should be considered as replicates.

> Response:

We will clarify the reference to "three proglacial areas" accordingly by changing this in the title to 'Primary succession and its driving variables - a cross-shere approach applied in the proglacial areas of the upper Martell Valley (Eastern Italian Alps)'. Moreover, in the study area description we will precisely state that we are dealing with two proglacial areas and one sampling of lateral moraine. As for the issue of replicates, we don't see it that way. The individual measurement points became glacier-free at different times and also underwent different developments. Therefore, they are real replicates. We will describe this in the text also in such a way.

Relevant change:
**We wrote in the study area description now:** Totally 65 plots (Fig. 2c) were sampled in 2019/2020 (used already for the analysis by Knoflach et al. 2021). They were located on the ground and lateral moraines of Fürkele- and Zufallferner as well as on lateral moraines of Langenferner the elevation gradient.

**7** Line 63: I think it should be useful to add the successional steps reported by Knoflach et al. (2021) in the Fig. 2

Response:
Thank you for the valuable suggestion. We will modify the figure and make the succession stages evident..

Relevant change:
**In section 2 Study area:** Figure 2 c was adjusted.

**8** Lines 64-69: "to test the following hypotheses: i) Many of the known potential explanatory variables are correlated….. ii) The most important explanatory variables for vegetation cover development include …. iii) Disturbances such as geomorphic disturbance and grazing/trampling reduce ….. iv) We expected that there are no single potential explanatory variables, and we will provide a better understanding of primary succession for prediction of future development.": the hypothesis iv) should be written in the same format of the others "no single potential expl. Variables are expected…."

Response:
We will reformulate it: (iv) no single potential explanatory variables are expected. With the four tested hypotheses we aim to provide a better understanding of primary succession for prediction of future development

Relevant change:
**In section introduction - we reformulated the hypotheses:** (i) Many of the known potential explanatory variables are correlated and can be summarised to a few numbers of components. (ii) It is not a single drivers used in literature that are decisive, but much more the interaction of all of them. (iii) Disturbances such as geomorphic disturbances and grazing/trampling reduce cover, species number, and thus also changes species composition. With the three tested hypotheses we aim to provide a better understanding of primary succession for prediction of future development.

**9** Line 67: better to explicit what do you mean with "climatic variables".

Response:
Thank you for your suggestion: we will concretise it - temperature and precipitation.

Relevant change:
**We reformulated the hypotheses:** (i) Many of the known potential explanatory variables are correlated and can be summarised to a few numbers of components. (ii) It is not a single drivers used in literature that are decisive, but much more the interaction of them. .....

**10** Line 74: it is not clear in this sentence if vegetation survey itself was performed by Knoflach et al. (2021) or only plot identification. It should be clearer.

| 442 | Response: |
| 443 | We will clarify it: The vegetation surveys were performed by Ramskogler and used for the analysis in |
| 444 | Knoflach et al. (2021). We did not only do a plot identification. |
| 445 | |
| 446 | Relevant change: |
| 447 | **In section 2 Study area**: ...The study area extends from 2367 m above sea level (a.s.l.) to 2881 m a.s.l. and |
| 448 | is NE-SW orientated. Totally 65 plots (Fig. 2c) were sampled in 2019/2020 (used already for the analysis by |
| 449 | Knoflach et al., 2021). |

**11** Line 93: repetition of "for primary succession"

| 452 | Response: |
| 453 | We will delete the repetition. |

**12** Lines 101-102 "we excluded variables only mentioned once or twice (e.g., wind exposure, snow depth, or soil type): It is not clear this criterion, since I would have said that snow depth could be strongly related to climate change.

| 458 | Response: |
| 459 | Of course, a variable that has hardly appeared as a driving variable in the literature so far can also make a |
| 460 | significant explanatory contribution. However, our approach was a compromise one: we focused on the |
| 461 | previous literature and included those variables that were mentioned more than 1-2 times. However, we did |
| 462 | not subject all other variables to in-depth analysis (which was not methodologically possible). However, we |
| 463 | assume, for example, that snow depth is very closely correlated with the number of snow free days and |
| 464 | therefore the essential information is also covered by this variable. |
| 465 | |
| 466 | Relevant change: |
| 467 | **In section 3.1 Literature review:** Definition of the potential explanatory variables: ...For example, it can be |
| 468 | assumed that snow depth is correlated with the snow free gdd, due to later melt out in places with higher |
| 469 | snow cover (Unterholzner et al., 2022). |
| 470 | **In section Discussion:** ...Snow free gdd can also be related to snow depth as with higher snow depth the melt |
| 471 | out will be later and thus we will have lower number of snow free gdd (Unterholzner et al., 2022). |

**13** Line 104: I would add a reference to the 31 explanatory variable list: (Tab1). It could be confusing to report in the text also the 39 variables found in literature, especially if you put in the text at first the table with only the 31 selected explanatory variables. Since, in another point, you report the 26 explanatory variables selected for PCA, the risk is that it become very confusing. Maybe, fig. 3 should be removed and the same information should be added in Tab1?

| 477 | Response: |
| 478 | Thank you for the valuable advice. We will take them up and rewrite the manuscript accordingly. We hope |
| 479 | that this will help us eliminate the confusion. |

| 480 | Relevant change: |
| 481 | **In section 3.1 Literature review:** Definition of the potential explanatory variables: We changed the table |
| 482 | and all variables are given in Table 1. Furthermore, an additional column was added to highlight which |
| 483 | variables were used in the analysis. |

**14** Line 118: I would change the title of the paragraph "3.2 Dependent variables: Vegetation indicators (Biosphere)" in "3.2 Dependent variables: Vegetation sampling (Biosphere)"

| 486 | Response: |
| 487 | We will change it. |
| 488 | |
| 489 | Relevant change: |
| 490 | 3.2. Dependent variables: Vegetation sampling (Biosphere) |

**15** Lines 122-124: "According to the change in species composition along the chronosequence, Knoflach et al. 2021
discriminated four successional stages: (i) a pioneer stage, (ii) an early successional stage, (iii) a late successional
stage with snowbed and grassland communities, and (iv) a climax stage with dwarf shrub - "I would make clear in
Fig. 2c the four successional stages.

Response:
Thank you, we will implement the succession stages.
Relevant change:
**In section 2 Study area:** Now the different successional stages are shown in Figure 2c. / In section 3.2.
Dependent variables: Vegetation sampling (Biosphere): …, and (iv) a dwarf shrub stage, by performing a
Nonmetric MultiDimensional Scaling (NMDS) and a Two Way INdicator SPecies ANalysis
(TWINSPNAN).
**16** Line 124: it is not a "climax" if the terrain deglaciated only 200 years ago

Response:
Climax will be replaced to dwarf shrub stage.
Relevant change:
**In section 3.2. Dependent variables:** Vegetation sampling (Biosphere): … , and (iv) a dwarf shrub stage,
by performing a Nonmetric MultiDimensional Scaling (NMDS) and a Two Way INdicator SPecies ANalysis
(TWINSPAN).
**17** Line 126: you did not consider any variables linked to arthropod succession. I think that in a "holistic approach"
this component should not be ignored along a glacier foreland: it is known, especially in pioneer stages, the importance
of arthropods as colonizer, even before plants appears. Then, their importance as disperser and pollinators could not
be ignored. In general, biosphere influences biosphere during succession and this point is not considered in the paper,
that is mainly focused in considering the impact of (mainly) physical factors on vegetation. Thus, I would not have
used the term "holistic".

Response:
Thank you for the very interesting suggestion. Unfortunately, we have no data about arthropods at all. But
we will include as a variable the ratio of competing (ccc, ccs, ccr) and non-competing species (all other
strategy types) in the analysis. In addition, we will also consider the proportion of life forms (grasses, forbs,
dwarf shrubs, lichens and mosses) as potential explanatory variables. Finally, we will also include the
availability of propagation material, the microclimatic effects of soil surface structure and the role of
arthropodes in the discussion.
Relevant change:
**Title:** Primary succession and its driving variables - a sphere-spanning approach applied in the proglacial
areas in the upper Martell Valley (Eastern Italian Alps). **In section 3.3: 3.3.2. Biosphere**
To take also the biosphere into account we used the Shannon-Index of the lifeforms, calculated from the
relative cover of the different lifeforms. For the different lifeforms the values were extracted from Landolt et
al. (2010). The csr-strategy types (Grime, 1974) were also extracted from Landolt et al. (2010). The species
were grouped to competitive species with two or three 'c' and not-competitive species (all other species). We
also included the relative cover of the not-competitive vascular plant species (Table1).
**18** Line 150: "of these ice-dammed lakes. (Fig. 1b)Further" delete dot before brackets, move it after brackets

Response:
We will move the dot after the brackets.

Relevant change:

**In section 2 Study area:** Figure 2 c was adjusted.

**19** Lines 149-151: are you sure that the succession restarted from zero? Organic matter should be present in soil after glacier lake outburst floods. I think you should better contextualized this point: have you checked the organic matter deposited and the grain size distribution?

Response:
We are sure as there is no significant difference in the soil parameters of these plots and similar plot not affected by the glacier lake outburst (affected plots: for humus [%] 3.62 (±0.44) in comparison to similar not affected plots humus [%] 2.61 (±0.25) did not differ significantly). This will be mentioned in the methodological section.

Relevant change:
**In section 3.3.3 Cryosphere:** ...There is no significant difference in the soil parameters of these plots and similar plot not affected by the glacier lake outburst (affected plots: for humus [%] 3.62 (±0.44) in comparison to similar not affected plots humus [%] 2.61 (±0.25)).

**20** Lines 152-153: "The parameter 'distance to the glacier front' was determined as the shortest distance from every single study plot to the glacier tongue using the 'near' function in ArcGIS 10.6". I would specify the year you are considering, even if it is guessable.

Response:
We will specify it: The glacier tongue extents are from the years when we did the surveys.

Relevant change:
**In section 3.3.3 Cryosphere:** The extent of the glacier toungues comes from the years when the according plots were surveyed.

**21** Line 157: insert dot after the reference.

Response:
We will do this.

**22** Line 158: "The distinction between no snow and snow cover was defined by a threshold of 5 mm snow water equivalent." Specify on which basis do you fix this threshold. Is it trustable?

Response:
The threshold of 5 mm SWE for the differentiation between snow and no snow coverage is commonly used in previous studies (e.g., Warscher et al. 2013, Brutel-Vuilmet et al. 2012, Najafi et al. 2016, Thorton et al. 2021, Conway et al. 2021, Hofmeister et al. 2022). However, the sensitivity of the threshold value is not often addressed. In the work of Hofmeister et al. (2022), two different SWE threshold values (i.e., 0 mm and 5 mm SWE) were evaluated against observed snow cover duration at one snow station. The 5 mm SWE threshold slightly outperformed the 0 mm threshold as it attained a slightly higher prediction accuracy. Conway et al. (2021) observed a smaller mean bias between modeled and observed snow cover duration when using a 5 mm threshold compared to 30 mm, which lead to a negative bias because the simulated snow cover duration is underestimated. We revised the sentence accordingly:"The distinction between no snow and snow cover was defined by a threshold of 5 mm snow water equivalent, which has been used in multiple studies (e.g., Warscher et al. 2013, Brutel-Vuilmet et al. 2012, Najafi et al. 2016, Thorton et al. 2021, Conway et al. 2021, Hofmeister et al. 2022)." Warscher et al. 2013 (DOI: 10.1002/wrcr.20219), Thorton et al. 2021 ( https://doi.org/10.1016/j.jhydrol.2021.126241), Brutel-Vuilmet et al. 2012 (doi:10.5194/tc-7-67-2013), Najafi et al. 2016 (https://doi.org/10.1007/s10584-016-1632-2), Conway et al. 2021 (DOI: 10.2307/27127990).

Relevant change:

**In section 3.3.3 Cryosphere:** ...The distinction between no snow and snow cover was defined by a threshold
of 5 mm snow water equivalent which has been used in multiple studies (e.g., Brutel-Vuilmet et al., 2013;
Najafi et al., 2016; Conway et al., 2021; Thornton et al., 2021; Hofmeister et al., 2022).
**23** Line 161: "TWI": since it is the first time TWI appears in the text I would explicit it: "Topographic wetness index
(TWI)"

Response:
We will do this.

Relevant change:
**In section 3.3.4 Hydrosphere:** The two hydrosphere-related variables were the precipitation and the
Topographic Wetness Index (TWI).

**24** Lines 183-185: why didn't you do soil analyzes of all the points to get direct values of some soil variables?

Response:
Due to financial constraints, we were unfortunately only able to sample a few sites in a first phase. In the
course of analysing the first results, however, we also realised that we needed concrete measurement data
for all sites. Therefore, we sampled all the vegetation plots in summer 2022 and since December we have
the results of the soil analyses from the lab. We will integrate the measured soil data into the new analysis.
Relevant change:
**In section 3.3.6 Pedosphere:** Soil analyses were performed on soil samples derived from three sampling
points (0-10 cm soil depth) for each of the study plots except the ones without soil development at the steep
lateral moraines. The samples were taken in August 2022. Only for soil moisture we used the community
weighted mean (m_w_) of the Landolt indicator value for soil moisture (F) (Landolt et al., 2010) was obtained
based on the single species cover on the plot. The suitability of indicator values as proxies for soil parameters
was described among others by e.g., Anschlag et al. (2017), Descombes et al. (2020), and Simon et al. (2020).
Soil samples were air-dried for one week and sieved afterwards up to 2 mm. Afterwards the soil samples
were analysed based on following methods: for pH - in CaCl2 (1:2.5), following  VDLUFA; sand, silt, and
clay were measured using the pipetting method according to the ÖNORM L1061-2;  humus [%], organic
carbon (C.org) [%], C:N ratio, and total nitrogen [%] following UNI EN 15936 (with a TOC-Analyser);
plant-available phosphorous (CAL-P [mg P2O5/100g] and plant-available potassium (CAL-K [mg
K2O/100g] using the Calcium-Acetat-Lactat-method following ÖNORM L 1087.
**25** Line 185 "for a subsample of the 65 study plots (n = 15)." How did you select this subsample? Which samples are
they?

Response:
We will specify it: The subsamples were only taken on less disturbed plots. We will remove this as we will
use the new data of the soil analysis and do not use the community weighted mean of the Landolt indicator
values anymore.
Relevant change:
We used now the soil samples taken in August 2022 (see above, comment 25).
**26** Lines 196-97: "Finally, the estimated cover of coarse-grained material (scree cover) in the field was used as an
additional independent variable (scree cover. "I would specify "for each plot".

Response:
We will specify this.
Relevant change:

**In section 3.3.6 Pedosphere:** ...Finally, the estimated cover of coarse-grained material (scree cover) for each plot was used as an addititonal independent variable (scree cover).

**27** line 200: the current signs of livestock grazing does not consider the effective influence in the past. If it is not possible to have information about past usage, it is better to clarify this point…

Response:
We have historical data of livestock density in this area. Instead of grazing (yes/no) we will use a max-standardised value for grazing density.

Relevant change:
**3.3.7 Anthroposphere:** ...To account for time, we calculated the max-standardised grazing intensity based on the number of animals per time period starting in 1869 (Supplement, Table S4)..

**28** Line 253: I would add "The most frequently analysed vegetation-related, dependent variables (biosphere)"

Response:
We can add "dependent variables ".

Relevant change:
**In section 4.1 Literature review:** Definition of the dependent and potential explanatory variables: The most frequently analysed vegetation-related, dependent variables (biosphere) in the literature …..

**29** Line 258: add ":" after "variables"

Response:
We will change it.

Relevant change:
**In section 3.3.3 Cryosphere:** ...There is no significant difference in the soil parameters of these plots and similar plot not affected by the glacier lake outburst (affected plots:  for humus [%] 3.62 (±0.44) in comparison to similar not affected plots humus [%] 2.61 (±0.25)).

**30** Table 3: why sometimes you use the name "RC1 etc" extrapolated from PCA and sometimes you use the full name of the variables?

Response:
Thank you for this comment. We will be consistent with the names.

Relevant change:
We added the names in Table 3. Furthermore, we used after introducing the names only those.

**31** Line 409-413: "In our study, we have now demonstrated that – contrary to our initial expectation – a series of other variables correlates with our hypothesised three variables, jointly described by the components RC1 ('elevation and time'), RC2 ('solar radiation'), RC3 ('south-eastness'), and RC5 '(low inclination'). Thus, hypothesis (ii) that the most important explanatory variables for vegetation cover development include years since deglaciation, elevation, and climatic variables, cannot be confirmed.": demonstrating that many variables are correlated also with years since deglaciation, elevation, and climatic variables does not means that they are less important. You should comment these results, even in relation to NMDS results, were there is clearly a pattern related to years since deglaciation.

Response:
The reviewer is right, of course. We will be happy to take this suggestion/note into account when rewriting the text.

Relevant change:
As the results changed due to including the soil parameters this part was rewritten.

**32** Line 424: delete "a"

Response:

**33** Line 440: "grazing and/or trampling showed no significant correlation" could be related to the fact that you did not considered grazing during all the period you considered (from LIA to now).

Response:
We have historical data of livestock density in this area. Instead of grazing (yes/no) we will use a max-standardised value for grazing density.

Relevant change:
**In section 5.2 Drivers for development of species number and species composition:** The stream power index (SPI) had a weak but significant negative effect on species number, grazing and/or trampling showed no significant correlation. For species composition we could show a significant effect of SPI as well as grazing/trampling by livestock.

**Public Comments**

**1** Dear authors, concerning the investigation of primary succession and related environmental variables, I suggest, if I may, checking the work by Garbarino et al. (2010) entitled "Patterns of larch establishment following deglaciation of Ventina glacier, central Italian Alps", published in Forest Ecology and Management. The paper focuses only on larch invasion in deglaciated areas in the forefield of Ventina glacier (Val Malenco, central Italian Alps) and tries to summarize several influencing factors of the phenomenon and can be considered, to a certain extent, a precursor of your research which copes with the issue in a broader context. Thanks for your kind attention.

Response:
Dear Daniolo Godone, of course we will look at the work by Garbarino et al. (2010) and also take notes on the drivers they mentioned. Some drivers could be similar or the same, but as we are analysing primary succession in the whole proglacial the drivers could also be different. Furthermore, we are also taking climate variables into account. Kind regards. Katharina Ramskogler.

Relevant change:
**In section discussion:** ...Another example that elevation matters was shown in the study of Garbarino et al. (2010) observed germination of larch between 14 to 34 years after deglaciation with denser tree stands at the lower sites. Their study area was lower in comparison to our study site, thus also the density of trees was higher. They also showed that facilitation did not matter for establishment of larch seedlings at their sites (Garbarino et al., 2010).